# InSAR Technique Applied to the Monitoring of the Qinghai-Tibet Railway

Qingyun Zhang[1,2], Yongsheng Li[2], Jingfa Zhang[2], Yi Luo[2]

[1]The first Monitoring and Application Center, China Earthquake Administration, Tianjin, China.
[2]Key Laboratory of Crustal Dynamics, Institute of Crustal Dynamics, China Earthquake Administration, 100085, Beijing

*Correspondence to*: Yongsheng Li (liyongsheng0217@163.com)

**Abstract.** The Qinghai-Tibet Railway is located on the Qinghai-Tibet Plateau and is the highest-altitude railway in the world. With the influence of human activities and geological disasters, it is necessary to monitor ground deformation along the Qinghai-Tibet Railway. In this paper, Advanced Synthetic Aperture Radar (ASAR) (T405 and T133) and TerraSAR-X data were used to monitor the Lhasa-Naqu section of the Qinghai-Tibet Railway from 2003 to 2012. The data period covers the time before and after the opening of the railway (total of ten years). This study used Full Rank Matrix Small Baseline Subset InSAR (FRAM-SBAS) time-series analysis to analyze the Qinghai-Tibet Railway. Before the opening of the railway (from 2003 to 2005), the Lhasa-Naqu road surface deformation was not obvious, with a maximum deformation of approximately 5 mm/yr; in 2007, the railway was completed and opened to traffic, and the resulting subsidence of the railway in the district of Damxung was obvious (20 mm/yr). After the opening of the railway (from 2008 to 2010), the Damxung segment included a considerable area of subsidence, while the northern section of the railway was relatively stable. The results indicate that FRAM-SBAS technology is capable of providing comprehensive and detailed subsidence information regarding railways with millimeter-level accuracy. An analysis of the distribution of geological hazards in the Damxung area revealed that the distribution of the subsidence area coincided with that of the geological hazards, indicating that the occurrence of subsidence in the Damxung area was related to the influence of surrounding geological hazards and faults. Overall, the peripheral surface of the Qinghai-Tibet Railway is relatively stable but still needs to be verified with real time monitoring to ensure that the safety of the railway is maintained.

## 1 Introduction

The Qinghai-Tibet Railway is located on the Qinghai-Tibet Plateau, with a total length of more than 1,100 km, of which 632 km is within a permafrost region. This railway is the highest-elevation railway in the world and the longest railway crossing over a permafrost region (Han et al., 2010). The key to success or failure of Qinghai-Tibet Railway lies in the treatment of frozen soil on subgrade. In particular, under global warming and the impacts of human activities and other factors, the stability of the railway roadbed in permafrost regions is facing great challenges (Wu et al., 2008; Liu et al., 2000; Wu et al., 2004). Permafrost is very sensitive to disturbance from external factors; the temperature increase decreases the strength of the frozen

soil, the bearing capacity of the frozen soil is reduced, and the ability to resist load is reduced (Wu et al., 2005). Moreover, the higher the ice content in the frozen soil is, the greater the extent of settlement after the frozen soil thaws. The Qinghai-Tibet Railway subgrade project adopted the design concept of active cooling (Cheng et al., 2003), based on the assumption of a stable substrate. During the construction of the Qinghai-Tibet Railway, the differential settlement and the countermeasures

employed in the road and bridge transition section in the permafrost region were studied in an experiment (Liu et al., 2004); countermeasures were proposed to address the differential settlement, but there are many bridges and the geological condition of the permafrost region is complex (Jin et al., 2008; Zhao et al., 2010). Therefore, the stability of the permafrost under the Qinghai-Tibet Railway is related to the normal operation and safety of railway.

In the past, investigations and monitoring of deformation of the Qinghai-Tibet Plateau mainly relied on field work (Welk,

1997; Brown et al., 2000; Ma et al., 2011), but the harsh natural conditions of certain areas of the Qinghai-Tibet Plateau led to heavy workloads and substantially increased the difficulty of performing traditional methods for measuring and analyzing deformation (Hu et al., 2007; Zhang et al., 2006). Therefore, it is necessary to explore a wide range of unattended, long-term, continuous methods for analyses of the deformation of the Qinghai-Tibet Railway. The development of interferometric synthetic aperture radar (InSAR) technology provides technical support for research on railway lifelines and transportation

networks.

Surface deformation monitoring is one of the most advantageous applications of InSAR technology. As far as the research and development of methods and techniques are concerned, InSAR technology has been improved from the use of a small amount of single-phase SAR data to analyzing time series and processing multiphase and multisource data. Differential InSAR (D-InSAR) technology has been developed on the basis of InSAR technology. Grabriel et al. (1989) first used D-InSAR

technology to monitor surface deformation, and the level of accuracy was maintained at the centimeter level. In subsequent decades, D-InSAR technology has been widely used in surface deformation, volcanic deformation, and seismic displacement monitoring. In view of the drawbacks of D-InSAR technology affected by the temporal baseline and spatial displacement, many scholars have proposed new technologies, such as InSAR time series analysis methods, to obtain better settlement detection results. The main time series analysis methods of InSAR include the Permanent Scatter Interferometry SAR (PS-

InSAR) method (Ferreti et al., 2000, 2001; Rosi et al., 2017; Yang et al., 2018), the Small Baseline Subset InSAR (SABS-InSAR) method (Berardino et al., 2002; Dong at al., 2014; Li et al., 2015; Corsetti et al., 2018) and the Stacking method (Lundgren et al., 2001; Chang et al., 2011; Dai et al., 2018). InSAR technology in China developed later but has also achieved good research results. Wang et al. (2000) used radar data to analyze the seismic deformation field and promote the development of D-InSAR technology in China. Li et al. (2009) improved the Atmospheric Phase Screen (APS) estimation algorithm on the

basis of the traditional InSAR short baseline analysis method and proposed the InSAR time series (TS) + atmospheric estimation model (AEM) method to obtain good application results. Li et al. (2015, 2016) proposed improved methods for identifying errors during phase unwrapping and correcting InSAR technology and advanced sequential InSAR analysis methods, and the study obtained very good results in practical applications.

Because the lifelines and transportation networks of railways and highways are long, linear projects, certain requirements must be met in the detection and analysis of such objects: First, large area coverage is necessary because the lifelines and transportation networks have different directions and unified reference data are needed for monitoring, requiring large-scale synchronous measurement. Second, the lifeline serves the people over a long time and requires long-term, near real-time

monitoring. The traditional geodetic deformation monitoring methods are mainly based on leveling and Global Positioning System (GPS) approaches. Traditional leveling is limited to discrete points on fixed routes. This approach is time consuming and expensive, so it is difficult to use for real-time monitoring. Although GPS technology can achieve high-precision monitoring, it is subject to the density of GPS points. The development of sequential InSAR technology provides a feasible technical means for real-time and long-term monitoring of long-standing linear engineering projects. At present, there are

many studies on the application of InSAR technology to lifelines and transportation networks (Hung et al., 2010; Shan et al., 2012; Chen et al., 2012; Qin et al., 2016; Hoope et al., 2014; Wang et al., 2017; Yu et al., 2013). The French SNCF Institute (2008) used TS-DInSAR technology to analyze the deformation along the Paris-Amsterdam high-speed railway and verified the feasibility of this method. Gatti et al. (2010) used PS-InSAR technology to analyze the deformation of a railway in Italy over three years, which further proved the applicability of time series InSAR technology. Daniel et al. (2018) used InSAR

technology to monitor the changes in highway bridges, which verified the feasibility of InSAR technology for monitoring lifelines and other facilities. Xing et al. (2018) improved the technical accuracy of PS-InSAR by installing angular reflectors, but the installation of angular reflectors requires manpower and materials and depends on the amount of SAR data generated by PS-InSAR technology.

At present, most of the lifeline monitoring is performed using PS-InSAR technology in order to overcome the disadvantage of

20 D-InSAR technology, which requires high data volume, and to improve the accuracy of deformation monitoring. In this paper, the InSAR method using the Full Rank Matrix (FRAM) Small Baseline Subset InSAR (SBAS) proposed by Li (2015) is used to study the deformation of the railway section from Lhasa to Naqu. Using Advanced Synthetic Aperture Radar (ASAR) and TerraSAR-X data, the crustal deformation information of the Qinghai-Tibet Railway over 10 years (from planning to construction of the railway) was obtained and the relationship between crustal deformation and surrounding geological

disasters was analyzed.

## 2 Study area

The Qinghai-Tibet Railway is a high-elevation railway that connects Xining (Qinghai Province) to Lhasa (Tibet Autonomous Region) (Fig. 1). The Qinghai-Tibet Railway and other national key projects that cross multiple active blocks and faults are

30 vulnerable to earthquakes and other disasters (Chen et al., 2018; Wu et al., 2016). Thus, monitoring the deformation of these projects is particularly important. InSAR and GPS are efficient techniques for monitoring the crustal deformation of Qinghai-Tibet blocks (Zhang et al., 2017).

批注 [z1]: Normative Map Name and Function Name

The Qinghai-Tibet Railway, highway, transmission line and other national key projects, with their ancillary studies, have the characteristics of strong correlations and continuous long-distance distributions. It is important to understand how to use these features to monitor the deformation of a long, linear region and reveal the movement of the Qinghai-Tibet Plateau block patterns with the deformations of these major project networks.

The Lhasa-Naqu section of the railway is located at the bottom of the southern valley of Nyainqentanglha Mountain in the central part of the Lhasa block (Jiang et al., 2018). In general, this section is north trending, and the Qinghai-Tibet Highway and Lhasa River pass through the area (Fig. 2). The terrain in the area is undulating, with the Nyainqentanglha Mountain Range in the northwest, a mountainous area in the southeast, and the Yangbajain-Damxung Basin in the middle of the region (Wu et al., 2018). The terrain is flat, the Qinghai-Tibet Railway and Qinghai-Tibet Highway pass through the basin, and the vegetation
along the railway is dense. Wetlands and low-lying regions are widely distributed, and the frozen soil in a long segment of the area contains abundant ice (Li et al., 2012). The study area is in a midlatitude region, and the land types mainly include glaciers, snow, bare rock and other land types. In this area, the Bengco fault lies across the railway; therefore, it is also important to study whether the movement of the Bengco fault affects the stability of the railway.

**3 Datasets and methodology**

**3.1 Datasets**

The greatest feature of the railway is its long linear engineering; to monitor and analyze this type of ground object, certain requirements must be met. The first is large area coverage because the road direction is not the same everywhere and, in monitoring, the need for a unified reference standard requires large-scale synchronized measurements. Second, to meet the requirements of InSAR technology, such projects must apply a high-precision InSAR data processing algorithm to ensure
high-resolution and fine-scale detection.

The TerraSAR-X data were acquired in stripmap mode with an incidence angle range of 39°-40° at HH polarization. The potential of the X-band data for detecting higher deformation gradients compared to that of other sensors arises from the high spatial and temporal resolutions of these data. Nevertheless, the coverage of the stripmap mode data is too small to study long, linear engineering projects. Therefore, in this paper, the C-band ASAR data and TerraSAR-X data were used in combination
to analyze the stability of the Qinghai-Tibet Railway. The TerraSAR-X data were selected to verify the accuracy of the ASAR T405 data results over the first segment of the railway in Yangbajain, and the ASAR T133 data were used to analyze the deformation of the railway near the Naqu area because the ASAR T405 data could not cover this area completely, and the ASAR T133 data could also verify the accuracy of the ASAR T405 data results over the Naqu area. The data coverage is shown in Figure 2 with the blue dotted line.
The ASAR T405 data were acquired from August 2003 to September 2010, but there were no data for 2006; therefore, we processed the data in three stages (2003-2005, 2007, and 2008-2010). The ASAR T133 data were acquired from November 2007 to August 2010. The TerraSAR-X data were acquired from December 2011 to November 2012.

批注 [z2]: Normative Map Name and Function Name

## 3.2 Methodology

The main steps of the FRAM-SBAS (Full Rank Matrix-Small Baseline Subset InSAR) method are as follows:

Firstly, the principle of interferogram generation is based on a specific time baseline and space baseline, and the appropriate redundant interferogram is selected to maximize the interferogram coherence. The main constraints are Eq. (1):

$$
\begin{aligned}
\left|\Delta B_{\perp}\right| &< B_{\perp thr} \\
\left|\Delta t\right| &< t_{thr} \\
\left|\Delta DC\right| &< DC_{thr}
\end{aligned}
\tag{1}
$$

where $\Delta B_{\perp}$ is the vertical baseline of data interference pairs, $\Delta t$ is the time baseline and $\Delta DC$ is the Doppler frequency difference.

To minimize the spatial and temporal decorrelations, we constructed a baseline network (Fig.3) using the following criteria: perpendicular baselines shorter than 200 m and a daytime interval baseline of less than 180 days. Each acquisition node in the network has at least two link pairs, meaning that each node has a minimum number of connections with the other nodes (two are used in this paper).

Secondly, coherence points are selected. The coherence point is selected based on the principle of full rank matrix, which effectively improves the quality of coherent point selection and provides the basis for subsequent least squares inversion. By constructing a single set interferogram network, each point can construct the matrix described by Equation 1, where $A$ is an $M*N$ dimensional matrix, $M$ is the number of interferograms, and $N$ is the number of images. For any pixel in any interferogram, the coherence is greater than a certain threshold, and the -1 and 1 flags can be set at the corresponding positions of the matrix A of Equation 1. For example, the first interferogram consists of the first image and the third image; that is, $\delta_{\phi 1} = \phi_3 - \phi_1$. If the interference of a certain point in the interferogram satisfies the conditions, then the corresponding position are $A_{11} = -1$ and $A_{13} = 1$, and the remaining positions of the first line are 0. Similarly, in the second interferogram, if the coherence of the point in one of the interferograms is less than the coherence threshold, the diversion is set to zero. All interferograms are considered to obtain each point pair matrix, and then the rank of each matrix is determined (Eq. (2)). If the matrix is full rank, the point is selected as the coherence point. The method can be used to select points that are coherent in the time series and coherent in the partial time interval but the interference network is connected, thereby increasing the number and precision of the coherent points.

$$A = \begin{bmatrix} -1 & 0 & 1 & \cdots & 0 & 0 & 0 \\ 0 & -1 & 1 & \cdots & 0 & 0 & 0 \\ & & \multicolumn{3}{c}{\dotfill} & & \\ & & \multicolumn{3}{c}{\dotfill} & & \\ 0 & 0 & 0 & \cdots & -1 & 1 & 0 \\ 0 & 0 & 0 & \cdots & -1 & 0 & 1 \end{bmatrix} \tag{2}$$

Thirdly, discrete point phase unwrapping is performed. In the FRAM-SBAS method, discrete coherence point data are resampled onto a regular Cartesian grid and phase unwrapping is performed using a network flow method. Then, the phase jump is checked according to the closed ring residual method, and the jump phase is corrected for the jump region.

Because of the change in the water vapor content in the atmosphere, phase artifacts in InSAR images caused by path delays, such as radar signal propagations through the stratified and turbulent atmosphere and ionosphere, frequently degrade the interpretability of the phase and correlation signatures of the terrain. The effect of atmospheric delay consists of three parts: 1) the long wavelength effect of the atmosphere, which is similar to the orbit error effect; 2) the short wavelength effect of the atmosphere (i.e., turbulent atmospheric artifacts); and 3) the vertical stratification of the atmosphere, which causes height-

dependent refractivity variations. In this paper, the three phase delays are calculated using a network methodology. The methodology estimates the phase delay for each SAR acquisition; then, each atmospheric artifact is simulated. The proposed method can effectively eliminate the atmospheric phase delay in the interferograms.

Fourthly, orbital and atmospheric error removal is performed. The orbit error removal is performed using the network method proposed by Biggs et al. (2007). The atmospheric error is divided into long-wavelength atmospheric delay error and turbulent

atmospheric delay error and terrain-related atmospheric delay error. The three errors are removed using the network methods. Traditional atmospheric delay phase (APS) estimates are based on a single interferogram (Ferretti et al., 2001). The atmospheric phase in the interferogram is the difference in atmospheric phase delay between the sub-image and the main image. If one of the two images is used to generate other interferograms, the phase delay signal on the image is also passed to the other interferograms, which also makes a correlation between the two interferograms. In this paper, we will use the network

method to estimate the atmospheric delay error of each image acquisition time and then use these estimates to obtain the delay error of a single moment to reconstruct the atmospheric delay error of each interferogram.

After removal of the DEM error and the deformation phase, it can be assumed that the residual phase is mainly caused by the atmosphere. Suppose $\delta\varphi_j(x,y)$ represents the residual phase value at $(x,y)$ on the $j$th interferogram and that $\varphi(t_A, x, y)$ and

$\varphi(t_B, x, y)$ represent the phase values of the imaging moments $t_A$ and $t_B$ at $(x,y)$, respectively. Each interferogram can

be expressed by Eq. (3):

$$\delta\varphi_j(x,y) = \varphi(t_B, x, y) - \varphi(t_A, x, y) \tag{3}$$

Based on a short baseline set network, we can construct equations such as Eq. (4):

$$\delta\varphi = A * \varphi \qquad (4)$$

where $A$ represents the $M * N$ matrix. The element $A_{kl}$ of the matrix $A$ is defined according to the following rules: If $l = t_B$, then $A_{kl} = 1$; if $l = t_A$, then $A_{kl} = -1$; otherwise, $A_{kl} = 0$. $\delta\varphi$ is a known vector of $M$ dimension, representing the number of interferograms $M$; $\varphi$ is an N-dimensional unknown vector representing the atmospheric phase values of $N$ imaging moments. Eq. (4) can be rewritten as follows:

$$\begin{bmatrix} \delta\varphi_1(x,y) \\ \vdots \\ \delta\varphi_k(x,y) \end{bmatrix} = \begin{bmatrix} -1 & 0 & 1 & & \\ & \ddots & & \ddots & \\ & & 0 & -1 & 1 \end{bmatrix} \begin{bmatrix} \varphi^{t_0}(x,y) \\ \vdots \\ \varphi^{t_k}(x,y) \end{bmatrix} \qquad (5)$$

where $\delta\varphi_k(x,y)$ represents the residual phase of interferogram $k$ and the corresponding position is (x, y).

Since the matrix $A$ is the rank-deficient matrix, a unique solution cannot be obtained. Generally, the singular value decomposition (SVD) method can be used to solve the solution and the atmospheric delay at each moment is obtained; then, the phase value of each interferogram is simulated by using Eq. (5). In the calculation of the variance of the residual phase of each interferogram, if the interferogram has the lowest atmospheric variance, the atmospheric phase of the interferogram is assumed to be zero. This constraint is added to Eq. (5) to calculate the atmospheric delay phase of all other image acquisition moments (Li et al., 2014).

Fifthly, the deformation result is obtained. The interference pattern is settled using the least squares method to obtain the deformation results of the study area.

The specific procedure is illustrated in Fig. 4.

## 4 Results and discussion

To remove the influence of far-field topography and Earth movement around the railway, the image was clipped to retain a certain area along the railway line. The SAR data before and after the opening of the railway were processed to obtain crustal deformation information along the railway.

### 4.1 InSAR results

The deformation information obtained by ASAR T405 during the construction of the Lhasa-Naqu section of the Qinghai-Tibet Railway from 2003 to 2005 is shown in Fig. 5 (A). The deformation of the study area is very small during this period, and the maximum deformation is approximately 5 mm/yr.

In 2007, the Lhasa-Naqu section of the Qinghai-Tibet Railway was functionally completed and opened to traffic. Fig. 5 (B) shows the deformation information of the line obtained by ASAR T405 in 2007. The area of the line is obviously variable compared with that before the opening of the railway. In the area circled by the elliptical red dotted line, the deformation is

large, with a maximum value of 20 mm/yr. The operation of the train has a certain impact on the railway, which may because the roadbed bears the weight of the train, causing the roadbed to become compacted and sink.

From 2008 to 2010, three years after the smooth operation of the Qinghai-Tibet Railway, the deformation of the line slowed compared with that in 2007 (Fig. 5 (C-D)). The main deformation area is consistent with the deformation zone in 2007, mainly

in the area of Damxung, which may be due to the geothermal exploitation in the area. A comparison of ASAR T405 and ASAR T133 reveals that the maximum deformation in this area is 15 mm/yr. However, the deformation in the northern part of the railway is relatively stable.

The overlap area between TerraSAR-X and ASAR is located above Yangbajain, and the deformation field obtained from the two datasets is analyzed. Fig. 6 (A) indicates that the ASAR acquired the deformation field in the region from 2008 to 2010.

There is an obvious uplift area at the corner of the railway from 2008 to 2010, and the maximum cumulative uplift is up to 7 mm/yr. Fig. 6 (B) shows the deformation field acquired by TerraSAR-X from 2011 to 2013, and the deformation of the uplift area at the corner is obviously smaller and tends to be stable, but there are two subsidence areas in the lower left corner and upper right corner, where the maximum subsidence is 10 mm/yr. The reasons for the subsidence area are analyzed by superposing geological hazards.

For the present analysis, a section of the railway in Damxung is selected (Fig. 7). In 2003-2005 (Fig. 7 (A)), the section of the road surface was basically stable. In 2007 (Fig. 7 (B)), the construction of the railway resulted in a region with a large area of settlement, where the maximum settlement rate was 20 mm/yr. In 2008-2010 (Fig. 7 (C)), the subsidence area was reduced, and the maximum settlement rate was 10 mm/yr; the surface tended to be stable.

Based on the results of the deformation field obtained for 2009-2010, some high-voltage towers along the Lhasa-Naqu railway

are analyzed (Fig. 8). The results show that different sections of the towers have different drops or lifts. The maximum lifting capacity of D306 is 17 mm, and the maximum settlement of D269 is 18 mm. Most of the larger variables are located at the corner of the railway and within a section of the line.

In this paper, the SAR data from 2003 to 2012 are analyzed. It is concluded that during the construction of the Qinghai-Tibet Railway, the linear variable along the railway was approximately 10 mm/yr. After the completion of the traffic, the linear

variable along the railway was 4-8 mm/yr. Li et al. (2012) used SBAS technology to analyze the ENVISAT ASAR data from 1997 to 2010 in the vicinity of Yangbajain-Damxung of the Qinghai-Tibet Railway. It was found that the settlement rate near the railway was 2 mm/yr, and the impact of frozen soil was approximately 10 mm/yr. Zhang et al. (2017) used Sentinel-1 data to analyze the deformation variables of the Qinghai-Tibet Railway during the period of 2014-2016 in the Qinghai-Tibet Plateau. It is concluded that the settlement rate of the Qinghai-Tibet Railway is approximately -10 mm/yr and the settlement rate of the

rail-stabilized area is approximately -5 mm/yr. Ma et al. (2011) and Dong et al. (2013) found that the overall settlement rate of the Qinghai-Tibet Railway subgrade is <10 mm/yr. At the junction of the fracture, we verified this finding with GPS, and the GPS result was highly consistent with the deformation field acquired by InSAR. Chen et al. (2012) used C- and L-band small baseline SAR interferometry to analyze the interaction between permafrost and infrastructure along the Qinghai-Tibet Railway, and the results showed surface motions along the embankment primarily in the range of - 20 to + 20 mm/yr.

批注 [z13]: Normative Map Name and Function Name

批注 [z14]: Normative Map Name and Function Name

批注 [z15]: Normative Map Name and Function Name

批注 [z16]: Normative Map Name and Function Name

批注 [z17]: Normative Map Name and Function Name

批注 [z18]: Normative Map Name and Function Name

批注 [z19]: Normative Map Name and Function Name

批注 [z20]: Normative Map Name and Function Name

## 4.2 Deformation and hazards

The Qinghai-Tibet Railway runs along the Qinghai-Tibet Plateau and its eastern margin. Under the influence of the uplift of the Qinghai-Tibet Plateau, the topography is generally high in the west and low in the east. The Qinghai-Tibet Railway passes through structural units, such as the Naqu orogenic belt and the Lhasa block. The climatic region of the Qinghai-Tibet Plateau has obvious vertical zoning characteristics, large temperature differences between winter and summer, and strong freeze-thaw weathering, which makes the geological environment along the Qinghai-Tibet Railway and adjacent areas sensitive and conducive to the development and occurrence of geological disasters.

In this paper, the upper Yangbajain railway section is selected and the geological hazard and deformation fields are superimposed (Fig. 9). The geological hazards occur mostly in areas with large deformation fields, which are also in the corner along the railway. There is basically no distribution of geological hazard points in the stable area of the deformation field, which shows that the stability of the surrounding geological structure is affected by railway construction and other factors, resulting in frequent geological hazards. At the same time, to ensure the safe and normal operation of the railway, it is necessary to perform long-term monitoring of the geological and geomorphological conditions around the railway to avoid casualties and economic losses.

## 4.3 Deformation and the Bengco fault

Garthwaite et al. (2013) obtained the deformation field within the Qinghai-Tibet Plateau by using large-scale InSAR technique. The present slip rate of the Bengco fault is 1-4+1 mm/yr. At 90.30°E, the slip rate is 1 mm/yr, which is 40 km west of the rupture zone of a 1951 earthquake. At 90.75°E, which is located in the western part of the rupture zone, the sliding rate reaches 4 mm/yr (Garthwaite et al., 2013). Ryder also used the InSAR method to estimate the deformation rate of the Bengco fault. The right-lateral slip deformation at 90.4°E is approximately $1 \pm 1$ mm/yr, and that at 90.9°E is approximately $4 \pm 1$ mm/yr. It is believed that this fracture deformation is mainly caused by the post-earthquake stick-slip effect of two earthquakes in 1951 and 1952 (Ryder et al., 2014). Taylor et al. studied a series of V-shaped conjugate tectonic belts using the InSAR technique. The present slip rates of several tectonic belts in the central and western regions are consistent with GPS results (Taylor et al., 2006).

Using the 32 ASAR image scenes between 2004 to 2010 (two adjacent tracks, T176 and T405) (Fig. 10), the same treatment method as described in Section 3.2 was used to obtain the deformation rate of the Bengco fault zone. Three profiles, A-A', B-B', and C-C', are selected in the deformation rate map to analyze the slip rate of the Bengco fault (Fig. 11). It can be seen from the results that the formation rate of the Bengco area is between 1 and 3 mm/yr. The rate in the eastern segment is approximately 2-3 mm/yr, and that in the western segment is approximately 1-2 mm/yr. The intersection of the A-A' section line and the fault essentially coincides with the intersection of the Qinghai-Tibet Railway and the fault-breaking fault zone. From the results of the section deformation rate, it can be seen that the fault slip has little effect on the overall deformation of the railway.

批注 [z21]: Normative Map Name and Function Name

批注 [z22]: Normative Map Name and Function Name

批注 [z23]: Normative Map Name and Function Name

**5 Conclusions**

We used FRAM SBAS technology to measure deformation rates in the Lhasa Naqu railway section using 49 ASAR image scenes and 17 TerraSAR-X SAR image scenes collected between August 2003 and November 2012. Due to the lack of data for 2006, we divided the data into five groups. The two datasets can provide detailed deformation information on the Qinghai-Tibet Railway. The main conclusions of this work can be summarized as follows.

(1) Before the opening of the railway, the Qinghai-Tibet Railway deformation was very small and the roadbed was considered stable. During the period of operation after the completion of the railway, the roadbed was relatively unstable during the first two or three years, and different subsidence and uplift zones appeared along the railway. During the later stage, with the accumulation of running time, the road surface gradually stabilized and the shape variations gradually decreased.

(2) The whole railway is relatively stable. The influence of frozen soil around the railway is affected by surface deformation, but, in general, this condition is not very serious. However, through the analysis of geological hazards and deformation fields, segments where the deformation field changes greatly are more likely to experience geological disasters, and geological disasters and abnormal deformations most commonly occur at the corner of the railway. Therefore, it is necessary to conduct real-time deformation and geological hazard monitoring along the Qinghai-Tibet Railway.

(3) To ensure the sustainable development of the Qinghai-Tibet Railway, the following analyses are urgently needed. First, the continuous monitoring of ground surface subsidence near the Qinghai-Tibet Railway and surrounding regions must be carried out using a geodetic survey (e.g., GPS and InSAR). Second, the distribution of geological hazards along the Qinghai-Tibet Railway and the regional geological structure need to be analyzed in detail. Finally, for the long-term regional monitoring of InSAR data, the separation of different orbital data and the processing of massive data are also key problems that need to be solved.

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

**Figure 1. Distribution map of main stations along Qinghai-Tibet Railway. Red circle is the starting station, green circle is the viewing platform station, yellow circle is the ordinary station, white circle is the unattended station. The charts B and C show that the Qinghai-Tibet Railway has been built and opened to traffic.**

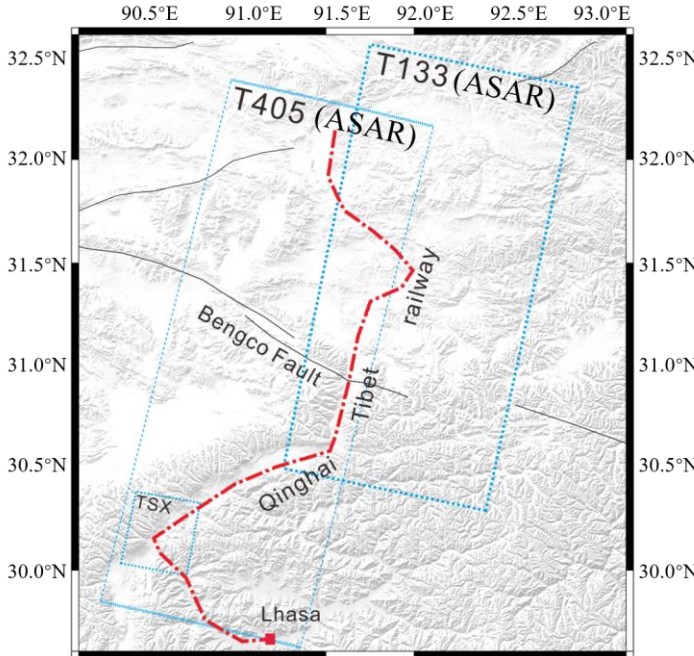

**Figure 2.** Study area and the distribution of the SAR imagery. The base map is derived from a digital elevation model (DEM). The blue dotted line shows the extents of the ASAR and TerraSAR-X images, the T405 and T133 represents the coverage of ASAR data, and the TSX represents the coverage of TerraSAR-X data. The red dotted line shows the railway. The black line shows the main fault in this area.

批注 [z24]: Unified map coordinates

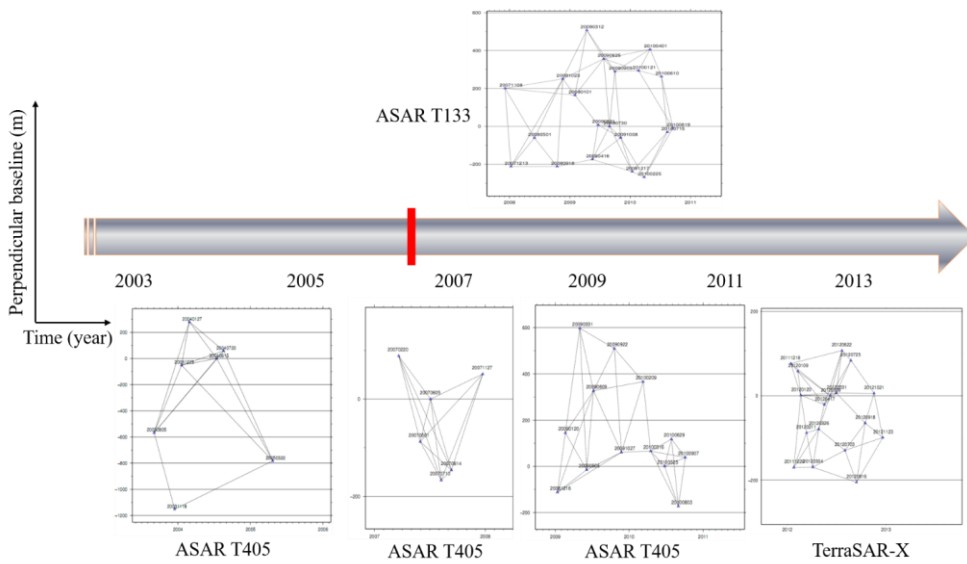

**Figure 3. Temporal and perpendicular baselines for the interferograms used in this study. Different graphs represent the baselines of different data generated during different time periods. In each figure, the horizontal axis represents the time (year) and the vertical axis represents the perpendicular baseline (m).**

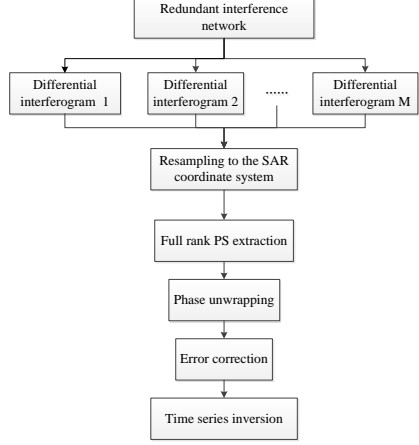

**Figure 4. Process flow of the Full Rank Matrix Small Baseline Subset InSAR (FRAM-SBAS) time-series analysis method**

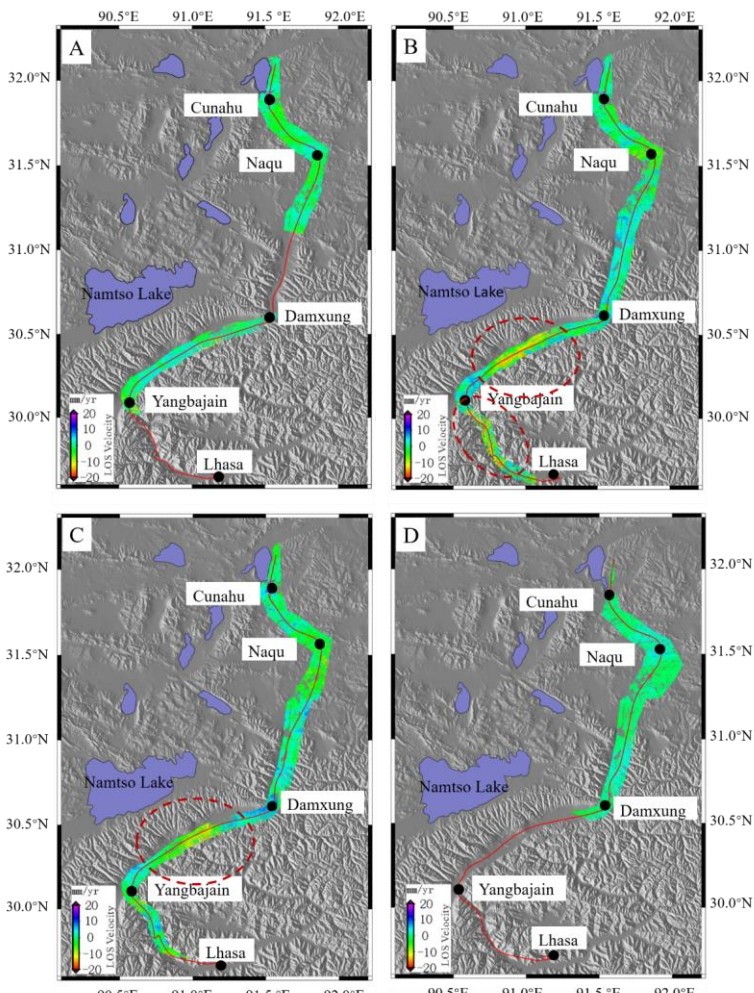

**Figure 5.** The InSAR results. Fig A represents the ASAR T405 results for the Lhasa-Naqu section of the Qinghai-Tibet Railway between 2003 and 2005. Fig B represents the ASAR T405 results for the Lhasa-Naqu section of the Qinghai-Tibet Railway in 2007, and the elliptical red dotted line represents the region of large deformation. Fig C represents the data are ASAR T405 data from 2008 to 2010. Fig D represents the data are ASAR T133 data from 2008 to 2010. The red line represents the railway. The black circle represents the main stations along the railway.

批注 [z25]: Unified map coordinates

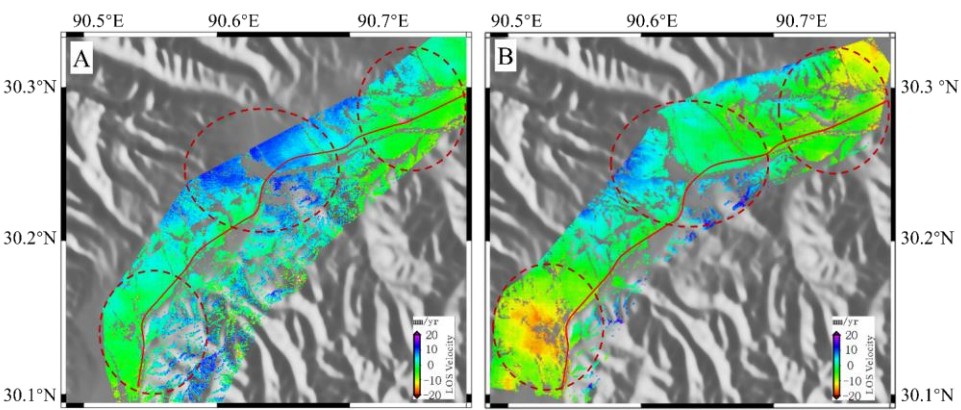

**Figure 6.** The deformation maps of the overlap area of the TerraSAR-X and ASAR data. (A) The ASAR T405 data range from 2008 to 2010. (B) TerraSAR-X data range from 2011 to 2012.

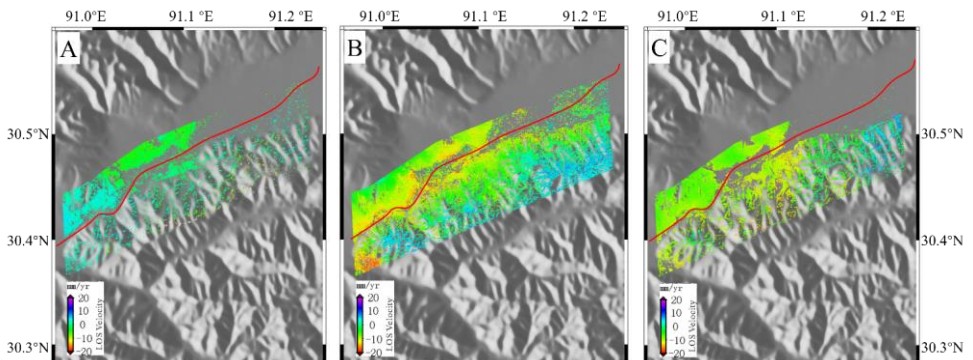

**Figure 7.** The deformation maps of the railway sequence changes from 2003 to 2010 in the Damxung section. **(A) 2003-2005; (B) 2007; (C) 2008-2010**

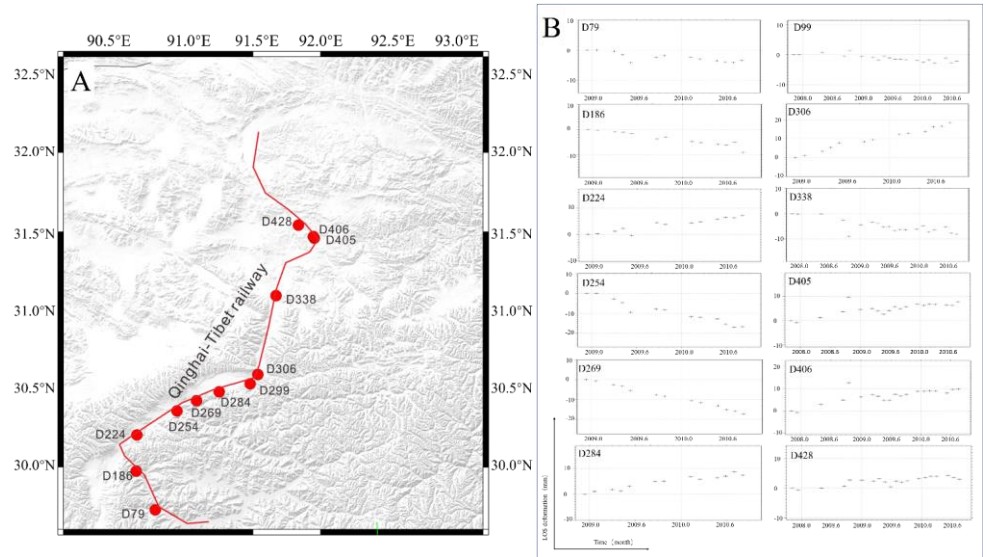

**Figure 8.** Deformation of a high-voltage power tower. (A) The location of the power tower. (B) The time series of the deformation characteristics of the power tower.

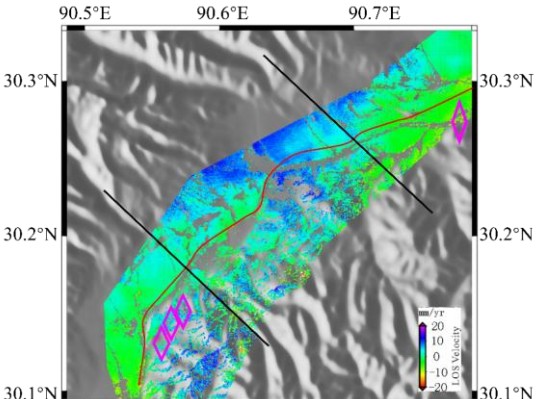

5    **Figure 9.** Deformation and hazards. The magenta rhombi show the geological hazard points.

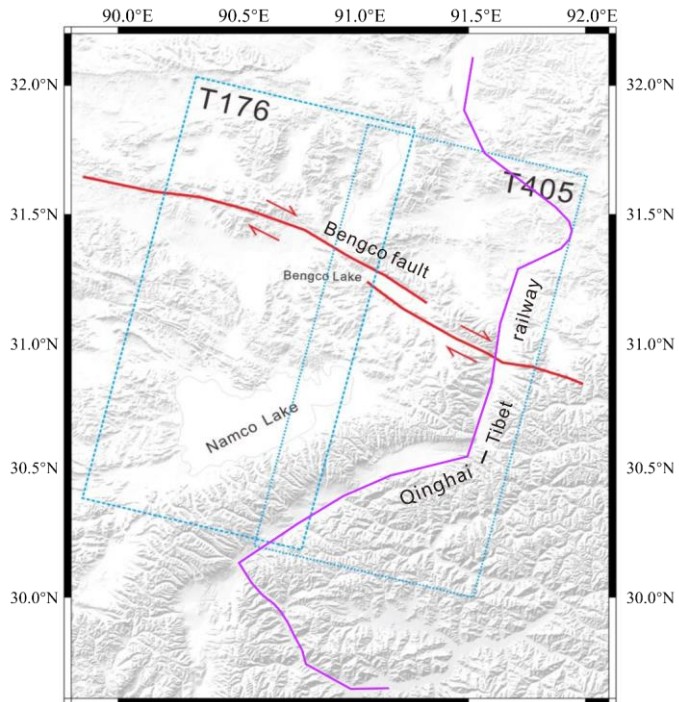

**Figure 10.** Bengco fault and the distribution of the SAR imagery. The blue dotted line shows the extents of the ASAR images. The red line shows the Bengco fault. The purple line shows the railway.

批注 [z30]: Unified map coordinates

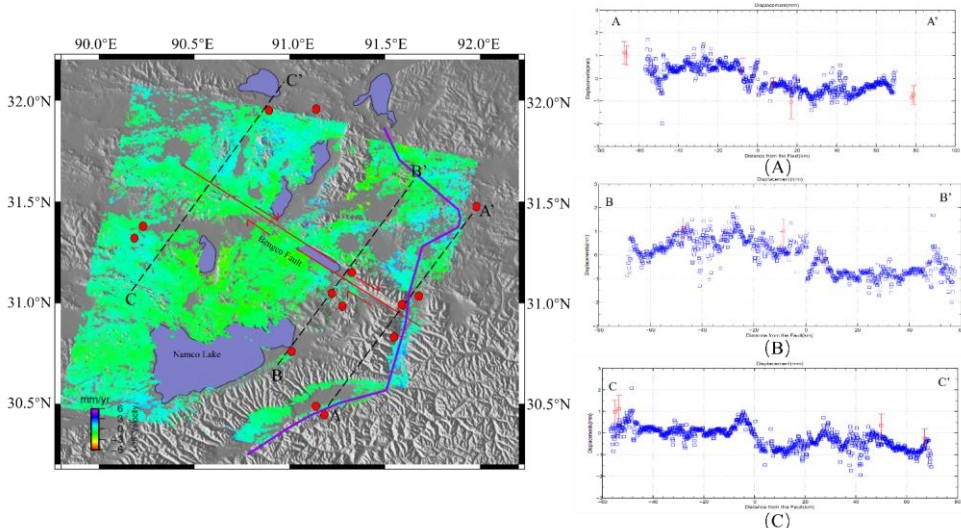

**Figure 11.** Bengco fault slip rate. The red line shows the Bengco fault. The purple line shows the railway. The red circle shows the GPS location. The black dotted line shows the profiles of the Bengco fault. The purple line shows the highway. The left figure shows the Bengco fault slip rate. The right figure shows the detailed information on profiles.

**Data availability.** The ENVISAT ASAR and TerraSAR-X data were obtained from the Dragon Programme IV (10607).

**Author contributions.** Qingyun Zhang designed the general idea and wrote the text content, Yongsheng Li was the technical director in chief, Jingfa Zhang and Yi Luo contributed to paper writing and revision. All authors discussed the editors' opinions and revised the paper.

**Competing interests.** The authors declare that they have no conflict of interest.

**Acknowledgements.** This work is supported by Research grants from National Natural Science Foundation of China under grant number 41704051 and Institute of Crustal Dynamics, China Earthquake Administration, under grant numbers ZDJ2018-16. We thank four anonymous reviewers for their detailed and thoughtful comments.

批注 [z31]: Unified map coordinates

批注 [z32]: Add Data availability and so on

**1、The main modifications:**

Unified map coordinates and Modify the figure and function name, add the Data availability and Author contributions and so on.