# Peer review of "InSAR Technique Applied to the Monitoring of the Qinghai-Tibet Railway"

_Natural Hazards and Earth System Sciences, 2018_

## Referee Comment (RC1) · Anonymous Referee #1 · 5 Dec 2018

General comments The authors used InSAR time series to investigate the deformation evolution of the Qinghai-Tibet Railway during 2003-2010. The topic will be of general interest to many.

Specific comments 1. The technique used in the data processing is the Full Rank Matrix (FRAM) Small Baseline Subset InSAR (SBAS) time-series analysis, but the method is not clear described in the main text. More details of the technique should be added in Section 3.2. 2. The deformation result from the study is not validated. And what is the precision or accuracy of the derived deformation?

Technical corrections 1. The reference style in P2/L1 and the following texts should be rewritten under the style of NHESSD, for example, "Xiaolei Lv et al., (2003)" should be changed to "Lv et al., (2003)". 2. P3/L7, global positioning systems (GPS) -> Global

Positioning Systems (GPS) 3. P3/L13-21, Need some references. 4. P3/L29-P4/9, The tense of these paragraphs should be the past tense. 5. In Figure 4, interferograms -> interferogram 6. The Figures 5, 6 and 7 can be combined as one figure.

---

## Referee Comment (RC2) · Anonymous Referee #2 · 2 Jan 2019

The paper is generally interesting. The innovative issues are mostly related to the application of one of the different variants of the SBAS algorithm to the case under investigation. This also represents the weak point of the paper. The applied method is mentioned but not summarized, and this makes the paper quite not self-consistent. The results are suitable. I would like to suggest the authors add a short description of the used methodology with a particular emphasis on the estimation and removal of APS. Some minor changes are required concerning English style. Reference to the literature is not adequate but it must be improved by searching for the most recent publications on the InSAR field. Also, the original SBAS paper of Berardino et al. has not been cited. Also, several other SBAS-like methods have been designed and presented in the literature. Generally, I suggest Chinese scholars not being, at most, concentrated

on Chinese publications, which are surely relevant and of great significance, but to open to worldwide literature.

---

## Author Comment (AC1) · 3 Jan 2019

**The comments of RC1**

**1、 Reply to the comment 1:**

**1.1 Comments from Referees**

The technique used in the data processing is the Full Rank Matrix (FRAM) Small Baseline Subset InSAR (SBAS) time-series analysis, but the methodis not clear described in the main text. More details of the technique should be added in Section 3.2.

**1.2 Author's response**

According to your suggestion, we will add the details of the technique in section 3.2. As followes:

The main key steps of the FRAM-SBAS (Full Rank Matrix-Small Baseline Subset InSAR) method are:

Firstly, the principle of interferogram generation is based on a specific time base and space baseline, and the appropriate redundant interferogram is selected to maximize the interferogram coherence. The main constraints are

$$\left|\Delta B_{\perp}\right| < B_{\perp thr}$$
$$\left|\Delta t\right| < t_{thr}$$
$$\left|\Delta DC\right| < DC_{thr}$$

Where $\Delta B_{\perp}$ is the vertical baseline of data interference pairs, $\Delta t$ is the time baseline and $\Delta DC$ is the Doppler frequency difference.

Secondly, selection of coherence points. The coherence point is selected based on the principle of full rank matrix, which effectively improves the quality of coherent point selection and provides the basis for subsequent least squares inversion. By constructing a single set interferogram network, each point can construct the following matrix (Equation 1). Where $A$ is a $M*N$ dimensional matrix, $M$ is the number of interferograms, and $N$ is the number of images. For any pixel in any interferogram, the coherence is greater than a certain threshold, and the -1 and 1 flags can be set at the corresponding positions of the matrix A of Equation 1. For example, the first interferogram consists of the first image and the third image, that is $\delta_{\phi 1} = \phi_3 - \phi_1$. If the interference of a certain point in the interferogram satisfies the condition, the corresponding position are $A_{11} = -1$ and $A_{13} = 1$, and the remaining positions of the first line are 0. Similarly, the second interferogram, if the coherence of the point in one of the

interferograms is less than the coherence threshold, the diversion is set to zero. All interferograms are judged to obtain each point pair matrix, and then the rank of each matrix is judged. If the matrix is full rank, the point is selected as the coherence point. The method can select coherent points that are coherent in the time series and coherent in the partial time interval but the interference network is connected, thereby increasing the number and precision of the coherent points.

$$A = \begin{bmatrix} -1 & 0 & 1 & \cdots & 0 & 0 & 0 \\ 0 & -1 & 1 & \cdots & 0 & 0 & 0 \\ & & \cdots\cdots\cdots\cdots\cdots\cdots \\ & & \cdots\cdots\cdots\cdots\cdots\cdots \\ 0 & 0 & 0 & \cdots & -1 & 1 & 0 \\ 0 & 0 & 0 & \cdots & -1 & 0 & 1 \end{bmatrix} \qquad (1)$$

Thirdly, discrete point phase unwrapping. In the FRAM-SBAS method, discrete coherence point data is resampled onto a regular Cartesian grid, and phase unwrapping is performed using a network flow method. Then, the phase jump is checked by the closed ring residual method, and the jump phase is corrected for the jump region.

Fourthly, orbital and atmospheric error removal. The orbit error removal is performed using the network method proposed by Biggs et al., (2007). The atmospheric error is divided into long-wavelength atmospheric delay error and turbulent atmospheric delay error and terrain-related atmospheric delay error. The three errors are removed using the network methods.

Traditional atmospheric delay phase (APS) estimates are based on a single interferogram (Ferretti et al., 2001). The atmospheric phase in the interferogram is the difference in atmospheric phase delay between the sub-image and the main image. If one of the two images is used to generate other interferograms, the phase delay signal on the image is also passed to the other interferograms, which also makes a correlation between the two interferograms. In this paper, we will use the network method to estimate the atmospheric delay error of each image acquisition time, and then use these estimates to obtain the delay error of a single moment to reconstruct the atmospheric delay error of each interferogram.

After removing the DEM error and the deformation phase, it can be assumed that the residual phase is mainly caused by the atmosphere. Suppose $\delta\varphi_j(x, y)$ represents the residual phase value at $(x, y)$ on the $j$th interferogram, and $\varphi(t_A, x, y)$ and $\varphi(t_B, x, y)$ represent the phase values of the imaging moments $t_A$ and $t_B$ at $(x, y)$, respectively. Each interferogram can be expressed by equation (2).

$$\delta\varphi_j(\mathrm{x,y}) = \varphi(t_B, x, y) - \varphi(t_A, x, y) \qquad (2)$$

Based on a short baseline set network, we can construct equations such as (3)

$$\boldsymbol{\delta\varphi = A * \varphi} \qquad (3)$$

Where $\boldsymbol{A}$ represents the $M * N$ matrix. The element $A_{kl}$ of the matrix $\boldsymbol{A}$ is defined according to the following rules: If $l = t_B$, then $A_{kl} = 1$; if $l = t_A$, then $A_{kl} = -1$; otherwise $A_{kl} = 0$. $\boldsymbol{\delta\varphi}$ is a known vector of $M$ dimension, representing the number of interferograms are $M$; $\boldsymbol{\varphi}$ is an N-dimensional unknown vector representing the atmospheric phase values of $N$ imaging moments. Equation (3) can be written as

$$\begin{bmatrix} \delta\varphi_1(x,y) \\ \vdots \\ \delta\varphi_k(x,y) \end{bmatrix} = \begin{bmatrix} -1 & 0 & 1 & & \\ & \ddots & & \ddots & \\ & & 0 & -1 & 1 \end{bmatrix} \begin{bmatrix} \varphi^{t_0}(x,y) \\ \vdots \\ \varphi^{t_k}(x,y) \end{bmatrix} \qquad (4)$$

Where $\delta\varphi_k(x,y)$ represents the residual phase of interferogram $k$, and the corresponding position is (x, y).

Since the matrix $\boldsymbol{A}$ is the rank-deficient matrix, a unique solution cannot be obtained. Generally, the singular value decomposition (SVD) method can be used to solve the solution, and the atmospheric delay at each moment is obtained, and then the phase value of each interferogram is simulated by using equation (4). By calculating the variance of the residual phase of each interferogram, if the interferogram has the lowest atmospheric variance, the atmospheric phase of the interferogram is assumed to be zero. This constraint is added to equation (4) to calculate the atmospheric delay phase of all other image acquisition moments (Li Yongsheng, 2014).

Fifthly, the deformation result is obtained. The interference pattern is settled using the least squares method to obtain the deformation results of the study area.

**1.3 Author's changes in manuscript.**

The contents of the response have been updated to the Section 3.2 (page 4 to page 6).

**2、 Reply to the comment 2:**

**2.1 Comments from Referees**

Comment 2: The deformation result from the study is not validated. And what is the precision or accuracy of the derived deformation?

**2.2 Author's response**

In Section 4.1, the comparison between the results of this paper and the results of others is added, and the reliability of the results is verified. As follows:

In this paper, the SAR data from 2003 to 2012 are analyzed. It is concluded that during the construction of the Qinghai-Tibet Railway, the linear variable along the railway is about 10mm/yr. After the completion of the traffic, the linear variable along the railway is 4~8mm/yr. Li et al., (2012) used used SBAS technology to analyze the ENVISAT ASAR data from 1997 to 2010 in the vicinity of Yangbajing-Dangxiong of Qinghai-Tibet Railway. It was found that the settlement rate near the railway was 2mm/yr, and the impact of frozen soil was about 10mm/yr. Zhang et al., (2017) used Sentinel-1 data to analyze the deformation variables of the Qinghai-Tibet Railway during the period of 2014-2016 in the Qinghai-Tibet Plateau. It is concluded that the settlement rate of the Qinghai-Tibet Railway is about -10 mm/yr and the settlement rate of the rail-stabilized area is about -5 mm/yr. Ma et al., (2011) and Dong et al., (2013) analyzed that the overall shape of the Qinghai-Tibet Railway subgrade is <10mm/yr. At the junction of the fracture, we verified it with GPS, which proves that the GPS result is highly consistent with the deformation field acquired by InSAR.

**2.3 Author's changes in manuscript.**

The contents of the response have been updated to the Section 4.1 (line 5-14 of page 7).

**3、 Reply to the comment 3:**

**3.1 Comments from Referees**

Comment 3: The reference style in P2/L1 and the following texts should be rewritten under the style of NHESSD, for example, "Xiaolei Lv et al., (2003)" should be changed to "Lv et al., (2003)".

**3.2 Author's response**

According to your suggestion, the references formats of the full text have been modified on the basis of your example.

**3.3 Author's changes in manuscript.**

The contents of the response have been updated to the full manuscript (page 1 to page 9).

**4、Reply to the comment 4:**

**4.1 Comments from Referees**

Comment 4: P3/L7, global positioning systems (GPS) -> Global Positioning Systems (GPS).

**4.2 Author's response**

According to your suggestion, we changed the global positioning systems (GPS) to Global Positioning Systems (GPS).

**4.3 Author's changes in manuscript.**

The contents of the response have been updated to the Section 2 (line 10 of page 3).

**5、Reply to the comment 5:**

**5.1 Comments from Referees**

Comment 5: P3/L13-21, Need some references.

**5.2 Author's response**

According to your suggestion, we will add references in section 2. As followes:

The Qinghai-Tibet Railway is a high-elevation railway that connects Xining (Qinghai Province) to Lhasa (Tibet Autonomous Region) (Figure 1). The Qinghai-Tibet Railway and other national key projects that cross multiple active blocks and faults are vulnerable to earthquakes and other disasters (Chen et al., 2018; Wu et al., 2016). Monitoring the deformation of these projects is of great significance. InSAR and Global Positioning Systems (GPS) are efficient techniques for monitoring the crustal deformation of Qinghai-Tibet blocks (Zhang et al., 2017).

The Qinghai-Tibet Railway, highway, transmission line and other national key projects, with their ancillary studies, have the characteristics of strong correlations and continuous long-distance distributions. We need to understand how to use these features to monitor the deformation of a long linear region and reveal the movement of the Qinghai-Tibet Plateau block patterns with the deformations of these major project networks.

The Lhasa-Nagqu part of the railway is located at the bottom of the southern valley of the Nyainqentanglha Mountain in the central part of the Lhasa block (Jiang et al., 2018). In general, it is

north-trending, and the Qinghai-Tibet highway and Lhasa River pass through the area. Figure 2 shows the study area, and the base map is derived from a digital elevation model (DEM). The terrain in the area is undulating, with the Nyainqentanglha Mountain Range in the northwest, a mountainous area in the southeast, and the Yangbajing-Damxung Basin in the middle of the region (Wu et al., 2018). The terrain is flat, the Qinghai-Tibet Railway and Qinghai-Tibet Highway pass through the basin, and the vegetation along the railway is rich. Wetlands and low-order regions are widely distributed, and the frozen soil in a long part of the area is rich with ice (Li et al., 2012). The study area is in a midlatitude region, and the land types mainly comprise glaciers, snow, bare rock and other land types. In this area, the Bengco fault lies across the railway; therefore, we also need to study whether the movement of the Bengco fault affects the stability of the railway.

**5.3 Author's changes in manuscript.**

The contents of the response have been updated to the Section 2 (line 7-25 of page 3).

**6、Reply to the comment 6:**

**6.1 Comments from Referees**

Comment 6: P3/L29-P4/9,The tense of these paragraphs should be the past tense.

**6.2 Author's response**

According to your suggestion, we changed the paragraph tense in section 3.1. As followes:

The TerraSAR-X data were acquired in stripmap mode, with an incidence angle range of 39 °-40 °at HH polarization. The potential of the X-band data for detecting higher deformation gradients compared to that of other sensors benefits from high spatial and temporal resolutions. Nevertheless, the coverage of the stripmap mode data was too small to study long linear engineering. Therefore, in this paper, the C-band ASAR data and TerraSAR-X data were used to analyze the stability of the Qinghai-Tibet Railway. The TerraSAR-X data was selected to verify the accuracy of the ASAR T405 data results over the first corner of the railway in Yangbajain and the ASAR T133 data were used to analyze the deformation of the railway near the Nagqu area because the ASAR T405 data could not cover this area completely, and the ASAR T133 data could also verify the accuracy of the ASAR T405 data results over the Nagqu area. The data coverage was shown in Figure 2 with the blue dotted line.

The ASAR T405 data were acquired from August 2003 to September 2010, but there were no data for 2016; therefore, we processed the data in three stages (2003-2005, 2007, and 2008-2010). The ASAR T133 data were acquired from November 2007 to August 2010. The TerraSAR-X data were acquired from December 2011 to November 2012.

**6.3 Author's changes in manuscript.**

The contents of the response have been updated to the Section 3.2 (line 3-14 of page 4).

**7、Reply to the comment 7:**

**7.1 Comments from Referees**

In Figure 4, interferograms -> interferogram

**7.2 Author's response**

According to your suggestion, we changed the interferograms to interferogram in figure 4. As follows:

[Figure]

**7.3 Author's changes in manuscript.**

The contents of the response have been updated to the figure 4 (page 15).

**8、Reply to the comment 8:**

**8.1 Comments from Referees**

Comment 8: The Figures 5, 6 and 7 can be combined as one figure.

**8.2 Author's response**

According to your suggestion, we combined the figure 5,6 and 7 as figure 5. As followes:

[Figure]

**8.3 Author's changes in manuscript.**

The contents of the response have been updated to the figure 5 (page 16).

**References**

Chen, T., Ma, W., and Zhou, G.: Numerical analysis of ground motion characteristics in permafrost regions along the Qinghai-Tibet Railway, Cold Regions Science & Technology, 148, 88-95, https://doi.org/10.1016/j.coldregions.2018.01.016, 2018.

Dong, C. H., and Zhao, X. Q.: Analysis on subgrade deformation features and influence factors in permafrost regions on Qinghai-Tibet Railway, Railway Standard Design, 6: 5-8, 2013.

Jiang, Y., Gao, Y., Dong, Z. B., Liu, B. L., and Zhao, L.: Simulations of wind erosion along the Qinghai-Tibet Railway in north-central Tibet, Aeolian Research, 32, 192-201, https://doi.org/10.1016/j.aeolia.2018.03.006, 2018.

Li, S. S.: The study of using SBAS to monitor the Motion of the frozen soil along Qinghai-Tibet railway, Central south university, 2012

Ma, W., Liu, D., and Wu, Q. B.: Monitoring and analysis of embankment deformation in permafrost regions of Qinghai-Tibet Railway, Rock Mechanics, 29(3) : 571-580, 2008.

Ma, W., Mu, Y. H., and Wu, Q. B.: Characteristics and mechanisms of embankment deformation along the Qinghai-Tibet Railway in permafrost regions, Cold Regions Science and Technology, 67(3) : 178-186, 2011.

Wu, Z. J., Ma, W., Chen, T., and Wang, L.: Dynamic Stability Analysis of Embankment Along the Qinghai-Tibet Railroad in Permafrost Regions, Environmental Vibrations and Transportation Geodynamics, 757-766, Doi: 10.1007/978-981-10-4508-070, 2016.

Zhang, Z. J.: Research on Qinghai-Tibet Permafrost Environment and Engineering using High Resolution SAR Images, Institute of Remote Sensing and Digital Earth, Chinese Academy of Science, 2017.

**The comments of RC2**

**1、Reply to the comment 1:**

**1.1 Comments from Referees**

Comment 1: I would like to suggest the authors add a short description of the used methodology with a particular emphasis on the estimation and removal of APS.

**1.2 Author's response**

According to your suggestion, we will add the details of the technique in section 3.2. As followes:

Traditional atmospheric delay phase (APS) estimates are based on a single interferogram (Ferretti et al., 2001). The atmospheric phase in the interferogram is the difference in atmospheric phase delay between the sub-image and the main image. If one of the two images is used to generate other interferograms, the phase delay signal on the image is also passed to the other interferograms, which also makes a correlation between the two interferograms. In this paper, we will use the network method to estimate the atmospheric delay error of each image acquisition time, and then use these estimates to obtain the delay error of a single moment to reconstruct the atmospheric delay error of each interferogram.

After removing the DEM error and the deformation phase, it can be assumed that the residual phase is mainly caused by the atmosphere. Suppose $\delta\varphi_j(x, y)$ represents the residual phase value at $(x, y)$ on the $j$th interferogram, and $\varphi(t_A, x, y)$ and $\varphi(t_B, x, y)$ represent the phase values of the imaging moments $t_A$ and $t_B$ at $(x, y)$, respectively. Each interferogram can be

expressed by equation (2).

$$\delta\varphi_j(x,y) = \varphi(t_B, x, y) - \varphi(t_A, x, y) \tag{2}$$

Based on a short baseline set network, we can construct equations such as (3)

$$\delta\varphi = A * \varphi \tag{3}$$

Where $A$ represents the $M * N$ matrix. The element $A_{kl}$ of the matrix $A$ is defined according to the following rules: If $l = t_B$, then $A_{kl} = 1$; if $l = t_A$, then $A_{kl} = -1$; otherwise $A_{kl} = 0$. $\delta\varphi$ is a known vector of $M$ dimension, representing the number of interferograms are $M$; $\varphi$ is an N-dimensional unknown vector representing the atmospheric phase values of $N$ imaging moments. Equation (3) can be written as

$$\begin{bmatrix} \delta\varphi_1(x,y) \\ \vdots \\ \delta\varphi_k(x,y) \end{bmatrix} = \begin{bmatrix} -1 & 0 & 1 & & \\ & \ddots & & \ddots & \\ & & 0 & -1 & 1 \end{bmatrix} \begin{bmatrix} \varphi^{t_0}(x,y) \\ \vdots \\ \varphi^{t_k}(x,y) \end{bmatrix} \tag{4}$$

Where $\delta\varphi_k(x,y)$ represents the residual phase of interferogram $k$, and the corresponding position is (x, y).

Since the matrix $A$ is the rank-deficient matrix, a unique solution cannot be obtained. Generally, the singular value decomposition (SVD) method can be used to solve the solution, and the atmospheric delay at each moment is obtained, and then the phase value of each interferogram is simulated by using equation (4). By calculating the variance of the residual phase of each interferogram, if the interferogram has the lowest atmospheric variance, the atmospheric phase of the interferogram is assumed to be zero. This constraint is added to equation (4) to calculate the atmospheric delay phase of all other image acquisition moments (Li Yongsheng, 2014).

**1.3 Author's changes in manuscript.**

The contents of the response have been updated to the Section 3.2 (line 1-25 of page 6).

**2、Reply to the comment 2:**

**2.1 Comments from Referees**

Comment 2: Some minor changes are required concerning English style.

**2.2 Author's response**

According to your suggestion, when the next manuscript is uploaded, the English style of the full text will be revised.

**2.3 Author's changes in manuscript.**

The contents of the response have been updated to the full-text.

**3、Reply to the comment 3:**

**3.1 Comments from Referees**

Comment 3: Reference to the literature is not adequate but it must be improved by searching for the most recent publications on the InSAR field. Also, the original SBAS paper of Berardino et al. has not been cited. Also, several other SBAS-like methods have been designed and presented in the literature.

**3.2 Author's response**

According to your suggestion, we will add the reference of the InSAR and SBAS. As followes:

In view of the drawbacks of D-InSAR technology affected by the temporal baseline and spatial displacement, many scholars have proposed new technologies such as InSAR Time Series Analysis Method, to obtain better settlement detection results. The main time series analysis methods of InSAR include Permanent Scatter Interferometry InSAR (PS-InSAR) method (Ferreti et al., 2000, 2001; Rosi et al., 2017; Yang et al., 2018), Small Baseline Subset InSAR (SABS-InSAR) method (Berardino et al., 2002; Dong at al., 2014; Li et al., 2015; Corsetti rt al., 2018) and Stacking method (Lundgren et al., 2001; Chang et al., 2011; Dai et al., 2018).

Reference:

Ferretti, A.: Nonlinear subsidence rate estimation using permanent scatters in differential SAR interferometry, IEEE Transactions on Geoscience & Remote Sensing, 38(5), 2202-2212, doi: 10.1109/36.868878, 2000.

Ferretti, A.: Permanent scatterers in SAR interferometry, IEEE Transactions on Geoscience & Remote Sensing, 39 (1), 8-20, doi: 10.1109/36.898661, 2001.

Rosi, A., Tofani, V., Tanteri, L., Tacconi, S. C., Agostini, A., Catani, F., and Casagli, N.: The new landslide inventory of Tuscany (Italy) updated with PS-InSAR: geomorphological features and landslide distribution, Landslides, 15(1), 5-19, doi: 10.1007/s10346-017-0861-4, 2017 .

Yang, C., Lu, Z., Zhang, Q., Zhao, C. Y., Peng, J. B., and Ji, L. Y.: Deformation at longyao ground fissure and its surroundings, north China plain, revealed by ALOS PALSAR PS-InSAR, International Journal of Applied Earth Observation and Geoinformation, 67, 1-9, https://doi.org/10.1016/j.jag.2017.12.010, 2018.

Berardino, P., Fornaro, G., Lanari, R., and Sansosti, E.: A New Algorithm for Surface Deformation Monitoring Based on Small Baseline Differential SAR Interferograms, Geoscience and Remote Sensing, IEEE Transactions, 40(11), 2375-2383, doi: 10.1109/TGRS.2002.803792, 2002.

Corsetti, M., Fossati, F., Manunta, M., and Marsella, M.: Advanced SBAS-DInSAR technique for controlling large civil infrastructures: An Application to the Genzano di Lucania Dam, Sensors, 18(7), doi: 10.3390/s18072371, 2018.

Dong, . (2014). Time-series analysis of subsidence associated with rapid urbanization in shanghai, china measured with sbas insar method. Environmental Earth Sciences, 72(3), 677-691.

Dong S. C., Samsonov, S., Yin, H. W., Ye, S. J., and Cao, Y. R.: Time-series analysis of subsidence associated with rapid urbanization in Shanghai, China measured with SBAS InSAR method, Environmental Earth Sciences, 72(3), 677-691, doi: 10.1007/s12665-013-2990-y, 2014.

Lundgren, P., Usai, S., Sansoti, E., Lanari, R., Tesauro, M., Fornaro G., and Berardino, P.: Modelling surface deformation observed with synthetic aperture radar interferometry at Campi Flegrei caldera, Journal of Geophysical Research, 106 (B9), 19355–19366, doi: 10.1029/2001jb000194, 2001.

Chang, Z. Q., Liu, X. M., Xue, T. F., and Yang, R. R.: Investigating ground subsidence in Beijing by using interferogram stacking InSAR, IEEE International Conference on Spatial Data Mining & Geographical Knowledge Services, IEEE, doi: 10.1109/ICSDM.2011.5969068, 2011.

Dai, K. R., Liu, G. X., Li, Z. H., Ma, D. Y., Wang, X. W., Zhang, B., Tang, J., and Li, G. Y.: Monitoring Highway Stability in Permafrost Regions with X-band Temporary Scatterers Stacking InSAR, Sensors, 18(6), 1-17, doi: 10.3390/s18061876, 2018.

**3.3 Author's changes in manuscript.**

The contents of the response have been updated to the Section 1 and the References (line 1-4 of page 2 and page 10-13).

---

## Author Response (AR1)

**The comments of RC1**

**1、Reply to the comment 1:**

**1.1 Comments from Referees**

The technique used in the data processing is the Full Rank Matrix (FRAM) Small Baseline Subset InSAR (SBAS) time-
series analysis, but the methodis not clear described in the main text. More details of the technique should be added in
Section 3.2.

**1.2 Author's response**

According to your suggestion, we will add the details of the technique in section 3.2. As follows:

The main steps of the FRAM-SBAS (Full Rank Matrix-Small Baseline Subset InSAR) method are as follows:

Firstly, the principle of interferogram generation is based on a specific time baseline and space baseline, and the appropriate
redundant interferogram is selected to maximize the interferogram coherence. The main constraints are

$$\left|\Delta B_{\perp}\right| < B_{\perp thr}$$
$$\left|\Delta t\right| < t_{thr} \qquad (1)$$
$$\left|\Delta DC\right| < DC_{thr}$$

where $\Delta B_{\perp}$ is the vertical baseline of data interference pairs, $\Delta t$ is the time baseline and $\Delta DC$ is the Doppler frequency
difference.

Secondly, coherence points are selected. The coherence point is selected based on the principle of full rank matrix, which
effectively improves the quality of coherent point selection and provides the basis for subsequent least squares inversion. By
constructing a single set interferogram network, each point can construct the matrix described by Equation 1, where $A$ is an
$M*N$ dimensional matrix, $M$ is the number of interferograms, and $N$ is the number of images. For any pixel in any
interferogram, the coherence is greater than a certain threshold, and the -1 and 1 flags can be set at the corresponding positions
of the matrix A of Equation 1. For example, the first interferogram consists of the first image and the third image; that is,
$\delta_{\phi 1} = \phi_3 - \phi_1$. If the interference of a certain point in the interferogram satisfies the conditions, then the corresponding position
are $A_{11} = -1$ and $A_{13} = 1$, and the remaining positions of the first line are 0. Similarly, in the second interferogram, if the
coherence of the point in one of the interferograms is less than the coherence threshold, the diversion is set to zero. All
interferograms are considered to obtain each point pair matrix, and then the rank of each matrix is determined. If the matrix is
full rank, the point is selected as the coherence point. The method can be used to select points that are coherent in the time
series and coherent in the partial time interval but the interference network is connected, thereby increasing the number and
precision of the coherent points.

$$A = \begin{bmatrix} -1 & 0 & 1 & \cdots & 0 & 0 & 0 \\ 0 & -1 & 1 & \cdots & 0 & 0 & 0 \\ & & \cdots\cdots\cdots\cdots\cdots\cdots & & & & \\ & & \cdots\cdots\cdots\cdots\cdots\cdots & & & & \\ 0 & 0 & 0 & \cdots & -1 & 1 & 0 \\ 0 & 0 & 0 & \cdots & -1 & 0 & 1 \end{bmatrix} \qquad (2)$$

Thirdly, discrete point phase unwrapping is performed. In the FRAM-SBAS method, discrete coherence point data are resampled onto a regular Cartesian grid and phase unwrapping is performed using a network flow method. Then, the phase jump is checked according to the closed ring residual method, and the jump phase is corrected for the jump region.

5  Fourthly, orbital and atmospheric error removal is performed. The orbit error removal is performed using the network method proposed by Biggs et al. (2007). The atmospheric error is divided into long-wavelength atmospheric delay error and turbulent atmospheric delay error and terrain-related atmospheric delay error. The three errors are removed using the network methods. Traditional atmospheric delay phase (APS) estimates are based on a single interferogram (Ferretti et al., 2001). The atmospheric phase in the interferogram is the difference in atmospheric phase delay between the sub-image and the main image.

10  If one of the two images is used to generate other interferograms, the phase delay signal on the image is also passed to the other interferograms, which also makes a correlation between the two interferograms. In this paper, we will use the network method to estimate the atmospheric delay error of each image acquisition time and then use these estimates to obtain the delay error of a single moment to reconstruct the atmospheric delay error of each interferogram.

After removal of the DEM error and the deformation phase, it can be assumed that the residual phase is mainly caused by the

15  atmosphere. Suppose $\delta\varphi_j(x, y)$ represents the residual phase value at $(x, y)$ on the $j$th interferogram and that $\varphi(t_A, x, y)$ and $\varphi(t_B, x, y)$ represent the phase values of the imaging moments $t_A$ and $t_B$ at $(x, y)$, respectively. Each interferogram can be expressed by Equation (3):

$$\delta\varphi_j(\text{x, y}) = \varphi(t_B, x, y) - \varphi(t_A, x, y) \qquad (3)$$

Based on a short baseline set network, we can construct equations such as (4):

20  $$\delta\boldsymbol{\varphi} = \boldsymbol{A} * \boldsymbol{\varphi} \qquad (4)$$

where $\boldsymbol{A}$ represents the $M * N$ matrix. The element $A_{kl}$ of the matrix $\boldsymbol{A}$ is defined according to the following rules: If $l = t_B$, then $A_{kl} = 1$; if $l = t_A$, then $A_{kl} = -1$; otherwise, $A_{kl} = 0$. $\delta\boldsymbol{\varphi}$ is a known vector of $M$ dimension, representing the number of interferograms $M$; $\boldsymbol{\varphi}$ is an N-dimensional unknown vector representing the atmospheric phase values of $N$ imaging moments. Equation (4) can be rewritten as follows:

$$\begin{bmatrix} \delta\varphi_1(x,y) \\ \vdots \\ \delta\varphi_k(x,y) \end{bmatrix} = \begin{bmatrix} -1 & 0 & 1 & & \\ & \ddots & & \ddots & \\ & & 0 & -1 & 1 \end{bmatrix} \begin{bmatrix} \varphi^{t_0}(x,y) \\ \vdots \\ \varphi^{t_k}(x,y) \end{bmatrix} \qquad (5)$$

where $\delta\varphi_k(x,y)$ represents the residual phase of interferogram $k$ and the corresponding position is (x, y).

Since the matrix $A$ is the rank-deficient matrix, a unique solution cannot be obtained. Generally, the singular value decomposition (SVD) method can be used to solve the solution and the atmospheric delay at each moment is obtained; then, the phase value of each interferogram is simulated by using Equation (5). In the calculation of the variance of the residual phase of each interferogram, if the interferogram has the lowest atmospheric variance, the atmospheric phase of the interferogram is assumed to be zero. This constraint is added to Equation (5) to calculate the atmospheric delay phase of all other image acquisition moments (Li et al., 2014).

Fifthly, the deformation result is obtained. The interference pattern is settled using the least squares method to obtain the deformation results of the study area.

**1.3 Author's changes in manuscript.**

The contents of the response have been updated to the Section 3.2 (page 5 to page 7).

**2、 Reply to the comment 2:**

**2.1 Comments from Referees**

Comment 2: The deformation result from the study is not validated. And what is the precision or accuracy of the derived deformation?

**2.2 Author's response**

In Section 4.1, the comparison between the results of this paper and the results of others is added, and the reliability of the results is verified. As follows:

In this paper, the SAR data from 2003 to 2012 are analyzed. It is concluded that during the construction of the Qinghai-Tibet Railway, the linear variable along the railway was approximately 10 mm/yr. After the completion of the traffic, the linear variable along the railway was 4-8 mm/yr. Li et al. (2012) used SBAS technology to analyze the ENVISAT ASAR data from 1997 to 2010 in the vicinity of Yangbajing-Dangxiong of the Qinghai-Tibet Railway. It was found that the settlement rate near the railway was 2 mm/yr, and the impact of frozen soil was approximately 10 mm/yr. Zhang et al. (2017) used Sentinel-1 data to analyze the deformation variables of the Qinghai-Tibet Railway during the period of 2014-2016 in the Qinghai-Tibet Plateau. It is concluded that the settlement rate of the Qinghai-Tibet Railway is approximately -10 mm/yr and the settlement rate of the rail-stabilized area is approximately -5 mm/yr. Ma et al. (2011) and Dong et al. (2013) found that the overall settlement rate of the Qinghai-Tibet Railway subgrade is <10 mm/yr. At the junction of the

fracture, we verified this finding with GPS, and the GPS result was highly consistent with the deformation field acquired by InSAR. Chen et al. (2012) used C- and L-band small baseline SAR interferometry to analyze the interaction between permafrost and infrastructure along the Qinghai-Tibet Railway, and the results showed surface motions along the embankment primarily in the range of - 20 to + 20 mm/yr.

**2.3 Author's changes in manuscript.**

The contents of the response have been updated to the Section 4.1 (page 8 to page 9).

**3、Reply to the comment 3:**

**3.1 Comments from Referees**

Comment 3: The reference style in P2/L1 and the following texts should be rewritten under the style of NHESSD, for example, "Xiaolei Lv et al., (2003)" should be changed to "Lv et al., (2003)".

**3.2 Author's response**

According to your suggestion, the references formats of the full text have been modified on the basis of your example.

**3.3 Author's changes in manuscript.**

The contents of the response have been updated to the full manuscript (page 1 to page 10).

**4、Reply to the comment 4:**

**4.1 Comments from Referees**

Comment 4: P3/L7, global positioning systems (GPS) -> Global Positioning Systems (GPS).

**4.2 Author's response**

According to your suggestion, we changed the global positioning systems (GPS) to Global Positioning Systems (GPS).

**4.3 Author's changes in manuscript.**

The contents of the response have been updated to the Section 2 (line 5 - 6 of page 3).

**5、Reply to the comment 5:**

**5.1 Comments from Referees**

Comment 5: P3/L13-21, Need some references.

**5.2 Author's response**

According to your suggestion, we will add references in section 2. As follows:

The Qinghai-Tibet Railway is a high-elevation railway that connects Xining (Qinghai Province) to Lhasa (Tibet Autonomous Region) (Figure 1). The Qinghai-Tibet Railway and other national key projects that cross multiple active blocks and faults are vulnerable to earthquakes and other disasters (Chen et al., 2018; Wu et al., 2016). Thus, monitoring the deformation of these projects is particularly important. InSAR and GPS are efficient techniques for monitoring the crustal deformation of Qinghai-

5 Tibet blocks (Zhang et al., 2017).

The Qinghai-Tibet Railway, highway, transmission line and other national key projects, with their ancillary studies, have the characteristics of strong correlations and continuous long-distance distributions. It is important to understand how to use these features to monitor the deformation of a long, linear region and reveal the movement of the Qinghai-Tibet Plateau block patterns with the deformations of these major project networks.

10 The Lhasa-Nagqu section of the railway is located at the bottom of the southern valley of Nyainqentanglha Mountain in the central part of the Lhasa block (Jiang et al., 2018). In general, this section is north trending, and the Qinghai-Tibet Highway and Lhasa River pass through the area. Figure 2 shows the study area, and the base map is derived from a digital elevation model (DEM). The terrain in the area is undulating, with the Nyainqentanglha Mountain Range in the northwest, a mountainous area in the southeast, and the Yangbajing-Damxung Basin in the middle of the region (Wu et al., 2018). The terrain is flat, the

15 Qinghai-Tibet Railway and Qinghai-Tibet Highway pass through the basin, and the vegetation along the railway is dense. Wetlands and low-lying regions are widely distributed, and the frozen soil in a long segment of the area contains abundant ice (Li et al., 2012). The study area is in a midlatitude region, and the land types mainly include glaciers, snow, bare rock and other land types. In this area, the Bengco fault lies across the railway; therefore, it is also important to study whether the movement of the Bengco fault affects the stability of the railway.

20 ## 5.3 Author's changes in manuscript.

The contents of the response have been updated to the Section 2 (page 3 to page 4).

**6、Reply to the comment 6:**

**6.1 Comments from Referees**

Comment 6: P3/L29-P4/9,The tense of these paragraphs should be the past tense.

25 ### 6.2 Author's response

According to your suggestion, we changed the paragraph tense in section 3.1. As follows:

The TerraSAR-X data were acquired in stripmap mode with an incidence angle range of 39°-40° at HH polarization. The potential of the X-band data for detecting higher deformation gradients compared to that of other sensors arises from the high spatial and temporal resolutions of these data. Nevertheless, the coverage of the stripmap mode data is too small to study long,

30 linear engineering projects. Therefore, in this paper, the C-band ASAR data and TerraSAR-X data were used in combination to analyze the stability of the Qinghai-Tibet Railway. The TerraSAR-X data were selected to verify the accuracy of the ASAR

T405 data results over the first segment of the railway in Yangbajain, and the ASAR T133 data were used to analyze the deformation of the railway near the Nagqu area because the ASAR T405 data could not cover this area completely, and the ASAR T133 data could also verify the accuracy of the ASAR T405 data results over the Nagqu area. The data coverage is shown in Figure 2 with the blue dotted line.

5 The ASAR T405 data were acquired from August 2003 to September 2010, but there were no data for 2016; therefore, we processed the data in three stages (2003-2005, 2007, and 2008-2010). The ASAR T133 data were acquired from November 2007 to August 2010. The TerraSAR-X data were acquired from December 2011 to November 2012.

**6.3 Author's changes in manuscript.**

The contents of the response have been updated to the Section 3.2 (page 4 to page 5).

10 ## 7、Reply to the comment 7:

**7.1 Comments from Referees**

In Figure 4, interferograms -> interferogram

**7.2 Author's response**

According to your suggestion, we changed the interferograms to interferogram in figure 4. As follows:

[Figure]

**7.3 Author's changes in manuscript.**

The contents of the response have been updated to the figure 4 (page 17).

**8、Reply to the comment 8:**

**8.1 Comments from Referees**

Comment 8: The Figures 5, 6 and 7 can be combined as one figure.

**8.2 Author's response**

5 According to your suggestion, we combined the figure 5,6 and 7 as figure 5. As follows:

[Figure]

**8.3 Author's changes in manuscript.**

The contents of the response have been updated to the figure 5 (page 18).

Zhang, Z. J.: Research on Qinghai-Tibet Permafrost Environment and Engineering using High Resolution SAR Images, Institute of Remote Sensing and Digital Earth, Chinese Academy of Science, 2017.

**The comments of RC2**

**1、Reply to the comment 1:**

20  ## 1.1 Comments from Referees

Comment 1: I would like to suggest the authors add a short description of the used methodology with a particular emphasis on the estimation and removal of APS.

**1.2 Author's response**

According to your suggestion, we will add the details of the technique in section 3.2. As follows:

25  Traditional atmospheric delay phase (APS) estimates are based on a single interferogram (Ferretti et al., 2001). The atmospheric phase in the interferogram is the difference in atmospheric phase delay between the sub-image and the main image. If one of the two images is used to generate other interferograms, the phase delay signal on the image is also passed to the other interferograms, which also makes a correlation between the two interferograms. In this paper, we will use the network method to estimate the atmospheric delay error of each image acquisition time and then use these estimates to obtain the delay

30  error of a single moment to reconstruct the atmospheric delay error of each interferogram.

After removal of the DEM error and the deformation phase, it can be assumed that the residual phase is mainly caused by the atmosphere. Suppose $\delta\varphi_j(x,y)$ represents the residual phase value at $(x,y)$ on the $j$th interferogram and that $\varphi(t_A,x,y)$ and $\varphi(t_B,x,y)$ represent the phase values of the imaging moments $t_A$ and $t_B$ at $(x,y)$, respectively. Each interferogram can be expressed by Equation (3):

$$\delta\varphi_j(\text{x},\text{y}) = \varphi(t_B,x,y) - \varphi(t_A,x,y) \tag{3}$$

Based on a short baseline set network, we can construct equations such as (4):

$$\boldsymbol{\delta\varphi} = \boldsymbol{A} * \boldsymbol{\varphi} \tag{4}$$

where $\boldsymbol{A}$ represents the $M*N$ matrix. The element $A_{kl}$ of the matrix $\boldsymbol{A}$ is defined according to the following rules: If $l=t_B$, then $A_{kl}=1$; if $l=t_A$, then $A_{kl}=-1$; otherwise, $A_{kl}=0$. $\boldsymbol{\delta\varphi}$ is a known vector of $M$ dimension, representing the number of interferograms $M$; $\boldsymbol{\varphi}$ is an N-dimensional unknown vector representing the atmospheric phase values of $N$ imaging moments. Equation (4) can be rewritten as follows:

$$\begin{bmatrix} \delta\varphi_1(x,y) \\ \vdots \\ \delta\varphi_k(x,y) \end{bmatrix} = \begin{bmatrix} -1 & 0 & 1 & & \\ & \ddots & & \ddots & \\ & & 0 & -1 & 1 \end{bmatrix} \begin{bmatrix} \varphi^{t_0}(x,y) \\ \vdots \\ \varphi^{t_k}(x,y) \end{bmatrix} \tag{5}$$

where $\delta\varphi_k(x,y)$ represents the residual phase of interferogram $k$ and the corresponding position is $(\text{x},\text{y})$.

Since the matrix $\boldsymbol{A}$ is the rank-deficient matrix, a unique solution cannot be obtained. Generally, the singular value decomposition (SVD) method can be used to solve the solution and the atmospheric delay at each moment is obtained; then, the phase value of each interferogram is simulated by using Equation (5). In the calculation of the variance of the residual phase of each interferogram, if the interferogram has the lowest atmospheric variance, the atmospheric phase of the interferogram is assumed to be zero. This constraint is added to Equation (5) to calculate the atmospheric delay phase of all other image acquisition moments (Li et al., 2014).

**1.3 Author's changes in manuscript.**

The contents of the response have been updated to the Section 3.2 (page 6 to page 7).

**2、 Reply to the comment 2:**

**2.1 Comments from Referees**

Comment 2: Some minor changes are required concerning English style.

**2.2 Author's response**

According to your suggestion, when the next manuscript is uploaded, the English style of the full text will be revised.

**2.3 Author's changes in manuscript.**

The contents of the response have been updated to the full-text.

**3、Reply to the comment 3:**

**3.1 Comments from Referees**

Comment 3: Reference to the literature is not adequate but it must be improved by searching for the most recent publications on the InSAR field. Also, the original SBAS paper of Berardino et al. has not been cited. Also, several other SBAS-like methods have been designed and presented in the literature.

**3.2 Author's response**

According to your suggestion, we will add the reference of the InSAR and SBAS. As follows:

In view of the drawbacks of D-InSAR technology affected by the temporal baseline and spatial displacement, many scholars have proposed new technologies, such as InSAR time series analysis methods, to obtain better settlement detection results. The main time series analysis methods of InSAR include the Permanent Scatter Interferometry SAR (PS-InSAR) method (Ferreti et al., 2000, 2001; Rosi et al., 2017; Yang et al., 2018), the Small Baseline Subset InSAR (SABS-InSAR) method (Berardino et al., 2002; Dong at al., 2014; Li et al., 2015; Corsetti et al., 2018) and the Stacking method (Lundgren et al., 2001; Chang et al., 2011; Dai et al., 2018).

**The minor issues of editor decision**

20  ## 1、Reply to the issue 1:

**1.1 Issues from editor decision**

Issue 1: please have a complete check of the English language by a native speaker

**1.2 Author's response**

According to your suggestion, when the next manuscript is uploaded, the English language was checked by a native
25  speaker.

**1.3 Author's changes in manuscript.**

The contents of the response have been updated to the full-text.

**2、 Reply to the issue 2:**

**2.1 Issues from editor decision**

Issue 2: please add more references related to previous case studies of the application of InSAR to the monitoring of lifelines and transportation networks. There are many of them in the recent literature and a comparison with your own work would be advisable.

**2.2 Author's response**

According to your suggestion, we will add more references related of the application of InSAR to the monitoring of lifelines and transportation networks in section 1. As follows:

Because the lifelines and transportation networks of railways and highways are long, linear projects, certain requirements must be met in the detection and analysis of such objects: First, large area coverage is necessary because the lifelines and transportation networks have different directions and unified reference data are needed for monitoring, requiring large-scale synchronous measurement. Second, the lifeline serves the people over a long time and requires long-term, near real-time monitoring. The traditional geodetic deformation monitoring methods are mainly based on leveling and Global Positioning System (GPS) approaches. Traditional leveling is limited to discrete points on fixed routes. This approach is time consuming and expensive, so it is difficult to use for real-time monitoring. Although GPS technology can achieve high-precision monitoring, it is subject to the density of GPS points. The development of sequential InSAR technology provides a feasible technical means for real-time and long-term monitoring of long-standing linear engineering projects. At present, there are many studies on the application of InSAR technology to lifelines and transportation networks (Hung et al., 2010; Shan et al., 2012; Chen et al., 2012; Qin et al., 2016; Hoope et al., 2014; Wang et al., 2017; Yu et al., 2013). The French SNCF Institute (2008) used TS-DInSAR technology to analyze the deformation along the Paris-Amsterdam high-speed railway and verified the feasibility of this method. Gatti et al. (2010) used PS-InSAR technology to analyze the deformation of a railway in Italy over three years, which further proved the applicability of time series InSAR technology. Daniel et al. (2018) used InSAR technology to monitor the changes in highway bridges, which verified the feasibility of InSAR technology for monitoring lifelines and other facilities. Xing et al. (2018) improved the technical accuracy of PS-InSAR by installing angular reflectors, but the installation of angular reflectors requires manpower and materials and depends on the amount of SAR data generated by PS-InSAR technology.

**2.3 Author's changes in manuscript.**

The contents of the response have been updated to the Section 1 and the References (line 1-18 of page 3 and page 11-15).

[revised manuscript text omitted]

5    **Figure 4. Process flow**

批注 [z11]: Reply to the comment 7 of RC1

[Figure]

批注 [z12]: Reply to the comment 8 of RC1

**Figure 5. The ASAR T405 results for the Lhasa-Naqu section of the Qinghai-Tibet Railway between 2003 and 2005. The red line represents the railway.**

[Figure]

**Figure 6. ASAR T405 results for the Lhasa-Naqu section of the Qinghai-Tibet Railway in 2007. The red line represents the railway.**
5  **The elliptical red dotted line represents the region of large deformation.**

[Figure]

**Figure 7. Deformation maps within three years after operation of the railway. (A) The data are ASAR T405 data from 2008 to 2010. (B) The data are ASAR T133 data from 2008 to 2010. The red line represents the railway.**

[Figure]

**Figure 8. The deformation maps of the overlap area of the TerraSAR-X and ASAR data. (A) The ASAR T405 data range from 2008 to 2010. (B) TerraSAR-X data range from 2011 to 2012.**

[Figure]

**Figure 9. The deformation maps of the railway sequence changes from 2003 to 2010 in the Damxung section.**

[Figure]

**Figure 10. Deformation of a high-voltage power tower. (A) The location of the power tower. (B) The sequence of the deformation characteristics of the power tower.**

[Figure]

5    **Figure 11. Deformation and hazards. The magenta rhombi show the geological hazard points.**

[Figure]

**Figure 12. Bengco fault and the distribution of the SAR imagery. The blue dotted line shows the extents of the ASAR images. The red line shows the Bengco fault. The purple line shows the railway.**

[Figure]

**Figure 13. Bengco fault slip rate. The red line shows the Bengco fault. The purple line shows the railway. The red circle shows the GPS location. The black dotted line shows the section lines of the Bengco fault. The purple line shows the highway. The left figure shows the Bengco fault slip rate. The right figure shows the detailed information on section lines.**

---

## Author Response (AR2)

[revised manuscript text omitted]

批注 [z6]: Reply to the "P4/L6-7, the description of base map should move to figure caption." of Referee #1

批注 [z7]: Reply to the "P5/L1, 2016 -> 2006" of Referee #1

**3.2 Methodology**

The main steps of the FRAM-SBAS (Full Rank Matrix-Small Baseline Subset InSAR) method are as follows:

Firstly, the principle of interferogram generation is based on a specific time baseline and space baseline, and the appropriate redundant interferogram is selected to maximize the interferogram coherence. The main constraints are

$$\begin{aligned} |\Delta B_{\perp}| &< B_{\perp thr} \\ |\Delta t| &< t_{thr} \\ |\Delta DC| &< DC_{thr} \end{aligned} \quad (1)$$

where $\Delta B_{\perp}$ is the vertical baseline of data interference pairs, $\Delta t$ is the time baseline and $\Delta DC$ is the Doppler frequency difference.

To minimize the spatial and temporal decorrelations, we constructed a baseline network (Figure 3) using the following criteria: perpendicular baselines shorter than 200 m and a daytime interval baseline of less than 180 days. Each acquisition node in the network has at least two link pairs, meaning that each node has a minimum number of connections with the other nodes (two are used in this paper).

Secondly, coherence points are selected. The coherence point is selected based on the principle of full rank matrix, which effectively improves the quality of coherent point selection and provides the basis for subsequent least squares inversion. By constructing a single set interferogram network, each point can construct the matrix described by Equation 1, where $A$ is an $M * N$ dimensional matrix, $M$ is the number of interferograms, and $N$ is the number of images. For any pixel in any interferogram, the coherence is greater than a certain threshold, and the -1 and 1 flags can be set at the corresponding positions of the matrix A of Equation 1. For example, the first interferogram consists of the first image and the third image; that is, $\delta_{\phi 1} = \phi_3 - \phi_1$. If the interference of a certain point in the interferogram satisfies the conditions, then the corresponding position are $A_{11} = -1$ and $A_{13} = 1$, and the remaining positions of the first line are 0. Similarly, in the second interferogram, if the coherence of the point in one of the interferograms is less than the coherence threshold, the diversion is set to zero. All interferograms are considered to obtain each point pair matrix, and then the rank of each matrix is determined. If the matrix is full rank, the point is selected as the coherence point. The method can be used to select points that are coherent in the time series and coherent in the partial time interval but the interference network is connected, thereby increasing the number and precision of the coherent points.

批注 [z8]: Reply to the "P5/L5-8, the paragraph can move to the end of L22." of Referee #1

$$A = \begin{bmatrix} -1 & 0 & 1 & \cdots & 0 & 0 & 0 \\ 0 & -1 & 1 & \cdots & 0 & 0 & 0 \\ & & & \cdots\cdots\cdots\cdots\cdots \\ & & & \cdots\cdots\cdots\cdots\cdots \\ 0 & 0 & 0 & \cdots & -1 & 1 & 0 \\ 0 & 0 & 0 & \cdots & -1 & 0 & 1 \end{bmatrix} \qquad (2)$$

Thirdly, discrete point phase unwrapping is performed. In the FRAM-SBAS method, discrete coherence point data are resampled onto a regular Cartesian grid and phase unwrapping is performed using a network flow method. Then, the phase jump is checked according to the closed ring residual method, and the jump phase is corrected for the jump region.

5    Because of the change in the water vapor content in the atmosphere, phase artifacts in InSAR images caused by path delays, such as radar signal propagations through the stratified and turbulent atmosphere and ionosphere, frequently degrade the interpretability of the phase and correlation signatures of the terrain. The effect of atmospheric delay consists of three parts: 1) the long wavelength effect of the atmosphere, which is similar to the orbit error effect; 2) the short wavelength effect of the atmosphere (i.e., turbulent atmospheric artifacts); and 3) the vertical stratification of the atmosphere, which causes height-

10   dependent refractivity variations. In this paper, the three phase delays are calculated using a network methodology. The methodology estimates the phase delay for each SAR acquisition; then, each atmospheric artifact is simulated. The proposed method can effectively eliminate the atmospheric phase delay in the interferograms.

批注 [z9]: Reply to the "P5/L9-15, the paragraph can move to P6/L11." of Referee #1

Fourthly, orbital and atmospheric error removal is performed. The orbit error removal is performed using the network method proposed by Biggs et al. (2007). The atmospheric error is divided into long-wavelength atmospheric delay error and turbulent

15   atmospheric delay error and terrain-related atmospheric delay error. The three errors are removed using the network methods. Traditional atmospheric delay phase (APS) estimates are based on a single interferogram (Ferretti et al., 2001). The atmospheric phase in the interferogram is the difference in atmospheric phase delay between the sub-image and the main image. If one of the two images is used to generate other interferograms, the phase delay signal on the image is also passed to the other interferograms, which also makes a correlation between the two interferograms. In this paper, we will use the network

20   method to estimate the atmospheric delay error of each image acquisition time and then use these estimates to obtain the delay error of a single moment to reconstruct the atmospheric delay error of each interferogram.

After removal of the DEM error and the deformation phase, it can be assumed that the residual phase is mainly caused by the atmosphere. Suppose $\delta\varphi_j(x,y)$ represents the residual phase value at $(x,y)$ on the $j$th interferogram and that $\varphi(t_A,x,y)$ and

$\varphi(t_B,x,y)$ represent the phase values of the imaging moments $t_A$ and $t_B$ at $(x,y)$, respectively. Each interferogram can

25   be expressed by Equation (3):

$$\delta\varphi_j(\mathrm{x,y}) = \varphi(t_B,x,y) - \varphi(t_A,x,y) \qquad (3)$$

Based on a short baseline set network, we can construct equations such as (4):

$$\delta\varphi = A * \varphi \tag{4}$$

where $A$ represents the $M*N$ matrix. The element $A_{kl}$ of the matrix $A$ is defined according to the following rules: If $l = t_B$, then $A_{kl} = 1$; if $l = t_A$, then $A_{kl} = -1$; otherwise, $A_{kl} = 0$. $\delta\varphi$ is a known vector of $M$ dimension, representing the number of interferograms $M$; $\varphi$ is an N-dimensional unknown vector representing the atmospheric phase values of $N$ imaging moments. Equation (4) can be rewritten as follows:

$$\begin{bmatrix} \delta\varphi_1(x,y) \\ \vdots \\ \delta\varphi_k(x,y) \end{bmatrix} = \begin{bmatrix} -1 & 0 & 1 & \\ & \ddots & & \ddots \\ & & 0 & -1 & 1 \end{bmatrix} \begin{bmatrix} \varphi^{t_0}(x,y) \\ \vdots \\ \varphi^{t_k}(x,y) \end{bmatrix} \tag{5}$$

where $\delta\varphi_k(x,y)$ represents the residual phase of interferogram $k$ and the corresponding position is (x, y).

Since the matrix $A$ is the rank-deficient matrix, a unique solution cannot be obtained. Generally, the singular value decomposition (SVD) method can be used to solve the solution and the atmospheric delay at each moment is obtained; then, the phase value of each interferogram is simulated by using Equation (5). In the calculation of the variance of the residual phase of each interferogram, if the interferogram has the lowest atmospheric variance, the atmospheric phase of the interferogram is assumed to be zero. This constraint is added to Equation (5) to calculate the atmospheric delay phase of all other image acquisition moments (Li et al., 2014).

Fifthly, the deformation result is obtained. The interference pattern is settled using the least squares method to obtain the deformation results of the study area.

The specific procedure is illustrated in Figure 4.

> 批注 [z10]: Reply to the "P7/L16-21, It can be omitted." of Referee #1

**4 Results and discussion**

To remove the influence of far-field topography and Earth movement around the railway, the image was clipped to retain a certain area along the railway line. The SAR data before and after the opening of the railway were processed to obtain crustal deformation information along the railway.

**4.1 InSAR results**

The deformation information obtained by ASAR T405 during the construction of the Lhasa-Naqu section of the Qinghai-Tibet Railway from 2003 to 2005 is shown in Figure 5 (A). The deformation of the study area is very small during this period, and the maximum deformation is approximately 5 mm/yr.

In 2007, the Lhasa-Naqu section of the Qinghai-Tibet Railway was functionally completed and opened to traffic. Figure 5 (B) shows the deformation information of the line obtained by ASAR T405 in 2007. The area of the line is obviously variable compared with that before the opening of the railway. In the area circled by the elliptical red dotted line, the deformation is

> 批注 [z11]: Reply to the "Figure 5, Figure 6 and Figure 7, what is size of the buffer? And the authors should label some city names." of Referee #1. Figures 6 and 7 were merged into Figure 5 before, but forgot to delete them. Therefore, after deletion, the figure numbers in the later part of the text have been changed and modified.

[revised manuscript text omitted]

批注 [z13]: Reply to the "P9/L12, what is meaning of "modified"?" of Referee #1

批注 [z14]: Reply to the "P9/L24, the reference is not complete." of Referee #1

批注 [z15]: Reply to the "P10/L2, section lines -> profiles" of Referee #1

批注 [z16]: Reply to the "P10/L3, the authors mentioned slip rate, but there is no further discussion on the value of slip rate." of Referee #1

[revised manuscript text omitted]

批注 [z18]: Reply to the "Figure 1, the caption is not self-explanation. Please revise it." of Referee #1

[Figure]

**Figure 2. Study area and the distribution of the SAR imagery. The base map is derived from a digital elevation model (DEM). The blue dotted line shows the extents of the ASAR and TerraSAR-X images, the T405 and T133 represents the coverage of ASAR data, and the TSX represents the coverage of TerraSAR-X data. The red dotted line shows the railway. The black line shows the main fault in this area.**

批注 [z19]: Reply to the "Figure 2, the coverage of ASAR and TerraSAR-X should be labeled." of Referee #1

[revised manuscript text omitted]

**5.3 Author's changes in manuscript.**

The contents of the response have been updated to the P3/L20.

**6、Reply to the comment 6:**

20  **6.1 Comments from Referees**

P4/L6-7, the description of base map should move to figure caption.

**6.2 Author's response**

According to your suggestion, we modified the statement and the figure name. As follows:

In general, this section is north trending, and the Qinghai-Tibet Highway and Lhasa River pass through the area (Figure 2).

**6.3 Author's changes in manuscript.**

The contents of the response have been updated to the P4/L7 and P16/figure 2.

**7、Reply to the comment 7:**

**7.1 Comments from Referees**

5  P5/L1, 2016 -> 2006

**7.2 Author's response**

According to your suggestion, we modified the statement. As follows:
The ASAR T405 data were acquired from August 2003 to September 2010, but there were no data for 2006;

**7.3 Author's changes in manuscript.**

10  The contents of the response have been updated to the P4/L30.

**8、Reply to the comment 8:**

**8.1 Comments from Referees**

P5/L5-8, the paragraph can move to the end of L22.

**8.2 Author's response**

15  According to your suggestion, we move the paragraph.

**8.3 Author's changes in manuscript.**

The contents of the response have been updated to the P5/L8-11.

**9、Reply to the comment 9:**

**9.1 Comments from Referees**

20  P5/L9-15, the paragraph can move to P6/L11.

**9.2 Author's response**

According to your suggestion, we move the paragraph.

**9.3 Author's changes in manuscript.**

The contents of the response have been updated to the P6/L5-L12.

**10、Reply to the comment 10:**

**10.1 Comments from Referees**

P7/L16-21, It can be omitted.

**10.2 Author's response**

According to your suggestion, we remove the paragraph.

**10.3 Author's changes in manuscript.**

The contents of the response have been updated to the P7/L16.

**11、Reply to the comment 11:**

**11.1 Comments from Referees**

P8/L3, "the crust of this section is relatively stable" is not formal. Please revise it.

**11.2 Author's response**

According to your suggestion, we modified the statement. As follows:

The deformation of the study area is very small during this period, and the maximum deformation is approximately 5 mm/yr.

**11.3 Author's changes in manuscript.**

The contents of the response have been updated to the P7/L23.

**12、Reply to the comment 12:**

**12.1 Comments from Referees**

P9/L12, what is meaning of "modified"?

**12.2 Author's response**

According to your suggestion, we modified the statement and remove "modified". As follows:
such as the Naqu orogenic belt and the Lhasa block.

**12.3 Author's changes in manuscript.**

The contents of the response have been updated to the P9/L4.

**13、Reply to the comment 13:**

**13.1 Comments from Referees**

5  P9/L24, the reference is not complete.

**13.2 Author's response**

According to your suggestion, we complete the reference. As follows:
Garthwaite et al. (2013) obtained the deformation field within the Qinghai-Tibet Plateau by using large-scale InSAR technique.

**13.3 Author's changes in manuscript.**

10  The contents of the response have been updated to the P9/L16.

**14、Reply to the comment 14:**

**14.1 Comments from Referees**

P10/L2, section lines -> profiles

**14.2 Author's response**

15  According to your suggestion, we change the section lines to profiles. As follows:
Three profiles, A-A', B-B', and C-C', are selected in the deformation rate map to analyze the slip rate of the Bengco fault (Figure 11).

**14.3 Author's changes in manuscript.**

The contents of the response have been updated to the P9/L26.

20  **15、Reply to the comment 15:**

**15.1 Comments from Referees**

P10/L3, the authors mentioned slip rate, but there is no further discussion on the value of slip rate.

**15.2 Author's response**

According to your suggestion, we have the discussion on the value of slip rate. As follows:

It can be seen from the results that the formation rate of the Bengco area is between 1 and 3 mm/yr. The rate in the eastern segment is approximately 2-3 mm/yr, and that in the western segment is approximately 1-2 mm/yr. The intersection of the A-A' section line and the fault essentially coincides with the intersection of the Qinghai-Tibet Railway and the fault-breaking fault zone. From the results of the section deformation rate, it can be seen that the fault slip has little effect on the overall deformation of the railway.

**15.3 Author's changes in manuscript.**

The contents of the response have been updated to the P9/L27-32.

**16、Reply to the comment 16:**

**16.1 Comments from Referees**

P10/L27, splicing -> separation

**16.2 Author's response**

According to your suggestion, we change the splicing to separation. As follows:
the separation of different orbital data and the processing of massive data are also key problems that need to be solved.

**16.3 Author's changes in manuscript.**

The contents of the response have been updated to the P10/L19.

**17、Reply to the comment 17:**

**17.1 Comments from Referees**

Figure 1, the caption is not self-explanation. Please revise it.

**17.2 Author's response**

According to your suggestion, we revise the caption of figure 1. As follows:
Distribution map of main stations along Qinghai-Tibet Railway. Red circle is the starting station, green circle is the viewing platform station, yellow circle is the ordinary station, white circle is the unattended station. The charts B and C show that the Qinghai-Tibet Railway has been built and opened to traffic.

**17.3 Author's changes in manuscript.**

The contents of the response have been updated to the P15/Figure 1.

**18、Reply to the comment 18:**

**18.1 Comments from Referees**

Figure 2, the coverage of ASAR and TerraSAR-X should be labeled.

**18.2 Author's response**

5 According to your suggestion, we change the caption of figure 2. As follows:

Figure 2. Study area and the distribution of the SAR imagery. The base map is derived from a digital elevation model (DEM). The blue dotted line shows the extents of the ASAR and TerraSAR-X images, the T405 and T133 represents the coverage of ASAR data, and the TSX represents the coverage of TerraSAR-X data. The red dotted line shows the railway. The black line shows the main fault in this area.

10 **18.3 Author's changes in manuscript.**

The contents of the response have been updated to the P16/Figure 2.

**19、Reply to the comment 19:**

**19.1 Comments from Referees**

15 Figure 3, the unit of y-axis is missing.

**19.2 Author's response**

According to your suggestion, we change the figure 3 and it's caption. As follows:

Figure 3. Temporal and perpendicular baselines for the interferograms used in this study. Different graphs represent the baselines of different data generated during different time periods. In each figure, the horizontal axis represents the time (year)
20 and the vertical axis represents the perpendicular baseline (m).

**19.3 Author's changes in manuscript.**

The contents of the response have been updated to the P17/Figure 3.

**20、Reply to the comment 20:**

**20.1 Comments from Referees**

25 Figure 4. the caption is not complete. Time sequence -> Time series

**20.2 Author's response**

According to your suggestion, we change the figure 4 and it's caption. As follows:

Process flow of the Full Rank Matrix Small Baseline Subset InSAR (FRAM-SBAS) time-series analysis method.

**20.3 Author's changes in manuscript.**

5 The contents of the response have been updated to the P17/Figure 4.

**21、Reply to the comment 21:**

**21.1 Comments from Referees**

Figure 5, Figure 6 and Figure 7, what is size of the buffer? And the authors should label some city names.

**21.2 Author's response**

10 According to your suggestion, we change the figure 4 and it's caption. As follows:

Figure 5. The InSAR results. Fig A represents the ASAR T405 results for the Lhasa-Naqu section of the Qinghai-Tibet Railway between 2003 and 2005. Fig B represents the ASAR T405 results for the Lhasa-Naqu section of the Qinghai-Tibet Railway in 2007, and the elliptical red dotted line represents the region of large deformation. Fig C represents the data are ASAR T405 data from 2008 to 2010. Fig D represents the data are ASAR T133 data from 2008 to 2010. The red line represents the railway.

15 The black circle represents the main stations along the railway.

Figures 6 and 7 were merged into Figure 5 before, but forgot to delete them. Therefore, after deletion, the figure numbers in the later part of the text have been changed and modified.

**21.3 Author's changes in manuscript.**

The contents of the response have been updated to the P7 to P9 and P19/Figure 5.

20 **22、Reply to the comment 22:**

**22.1 Comments from Referees**

Figure 10, sequence -> time series

**22.2 Author's response**

According to your suggestion, we change sequence to time series. As follows:

25 Figure 8. Deformation of a high-voltage power tower. (A) The location of the power tower. (B) The time series of the deformation characteristics of the power tower.

And because the delate of other figure, the number of this figure has changed.

**22.3 Author's changes in manuscript.**

The contents of the response have been updated to the P20/Figure 8.

**23、Reply to the comment 23:**

**23.1 Comments from Referees**

Figure 13, section lines -> profiles

**23.2 Author's response**

According to your suggestion, we change section lines to profiles. As follows:
Figure 11. Bengco fault slip rate. The red line shows the Bengco fault. The purple line shows the railway. The red circle shows the GPS location. The black dotted line shows the profiles of the Bengco fault. The purple line shows the highway. The left figure shows the Bengco fault slip rate. The right figure shows the detailed information on profiles.
And because the delate of other figure, the number of this figure has changed.

**23.3 Author's changes in manuscript.**

The contents of the response have been updated to the P22/Figure 11.

**24、Other modifications:**

The main place names of the text and the chart are unified. Nagqu changed to Naqu, Yangbajing changed to Yangbajain, Dangxiong changed to Damxung. The main place names modified in the text were marked red.